

# Uncertainty analysis of single- and multiple-size-class frazil ice models

Fabien Souillé[1], Cédric Goeury[1], and Rem-Sophia Mouradi[2,3]

[1]EDF R&D, National Laboratory for Hydraulics and Environment (LNHE), 6 Quai Watier, 78400 Chatou, France
[2]EDF R&D, Fluid Dynamics, Energy and Environment Department (MFEE), 6 Quai Watier, 78400 Chatou, France
[3]CEREA (Centre d'Enseignement et de Recherche en Environnement Atmosphérique), Joint Laboratory École des Ponts ParisTech and EDF R&D, Université Paris-Est, 77455 Marne-la-Vallée, France

**Correspondence:** Fabien Souillé (fabien.souille@edf.fr)

**Abstract.** The formation of frazil ice in supercooled waters has been extensively studied, both experimentally and numerically, in recent years. Numerical models, with varying degrees of complexity, have been proposed; these are often based on many parameters, the values of which are uncertain and difficult to estimate. In this paper, an uncertainty analysis of two mathematical models that simulate supercooling and frazil ice formation is carried out within a probabilistic framework. The two main

goals are (i) to provide quantitative insight into the relative importance of contributing uncertain parameters, to help identify parameters for optimal calibration, and (ii) to compare the output scatter of frazil ice models with single and multiple crystal size classes. The derivation of single- and multi-class models is presented in light of recent work, their numerical resolution is discussed, and a list of the main uncertain parameters is proposed. An uncertainty analysis is then carried out in three steps. Parameter uncertainty is first quantified, based on recent field, laboratory and numerical studies. Uncertainties are then

propagated through the models using Monte Carlo simulations. Finally, the relative influence of uncertain parameters on the output time series - i.e. the total frazil volume fraction and water temperature - is assessed by means of Sobol indices. The influence of input parameters on the long-term asymptotic as well as short-term transient evolution of the systems is discussed, depending on whether gravitational removal is included or not in the models.

## 1 Introduction

Formation of frazil ice in water bodies has been widely investigated because of its impact on submerged structures (Daly, 1991, 2006; Richard and Morse, 2008) and because it often precedes formation of ice cover in rivers and oceans (Daly, 1994; Smedsrud and Jenkins, 2004). For these reasons, the study of frazil formation processes is an active area of research, with a large variety of applications in river and coastal engineering.

The main drivers for the formation of frazil are the water cooling rate resulting from heat exchanges with the atmosphere,

the initial seeding of frazil nuclei and the turbulent mixing. Then, the thermal growth process deriving from the heat exchange between water and primitive crystals, allows a fine description of the balance between growth frazil ice and water supercooling. Previous mathematical models describing the evolution of frazil ice and water temperature, were based on ideas pioneered by Daly (1984, 1994). Omstedt (1985) developed a model based on a turbulent channel-flow boundary-layer theory, in which a





mean particle is considered of constant geometric properties and constant Nusselt number is considered. Mean particle-size
models have been incorporated in numerical hydraulic tools such as CRISSP2D (Shen and Wasantha Lal, 1993; Shen et al.,
1995; Shen, 2010). More complex models are based on the ice-number continuity equation introduced by Daly (1984), taking
into account the complexity of crystal distribution. In contrast with single-size-class models, the crystal number continuity
equations allow introduction of secondary nucleation and flocculation processes. When combined with thermal growth, these
complement the modeling of frazil crystal size evolution. Svensson and Omstedt (1994) proposed a numerical model for
solving the equations introduced by Daly (1984) by considering discrete radius intervals. Conceptual models for secondary
nucleation and flocculation were introduced and calibrated to fit the experimental data of Michel (1963) and Carstens (1966).
Hammar and Shen (1991) derived a variable particle-size model in two dimensions, and later improved it with secondary
nucleation and flocculation (Hammar and Shen, 1995). Variable particle-size models has also been integrated in TELEMAC-
MASCARET (Souillé et al., 2020). Numerous simplified implementations have been made that consider a well-mixed water
body. For example, Wang and Doering (2005) worked with the same model as Svensson and Omstedt (1994) and further
discussed calibration of influential parameters in initial seeding, secondary nucleation, and flocculation by comparing them
with the experimental data of Clark and Doering (2004). Implementation of multiple size class frazil dynamics in the context
of sea ice and ice shelf water plumes has also been proposed by Smedsrud (2002), Smedsrud and Jenkins (2004) and Holland
and Feltham (2005). In the same context, Rees Jones and Wells (2018) have recently shed new light on multiple-size-class
models and discussed crystal growth rate, and the occurrence of frazil explosion. They also identified and characterized steady
state crystal size distribution. Henceforth, single- and multiple-size-class models will be referred to as (SSC)[1] and (MSC)[2]
models respectively.

Common to all numerical model of frazil dynamics is their use of a large number of parameters, making calibration difficult.
The modeling studies mentioned above show that frazil ice models can be fitted to reproduce evolution of temperature and frazil
volume fraction. However, given the uncertainty of fitting parameters, it is questionable whether these models are predictive.
As Rees Jones and Wells (2018) point out, the consistency between experiments and models does not necessarily means
parameterization is correct. This is particularly true when different processes compete and have similar impact on the crystal
size distribution. Besides, introducing new processes in the models increases the number of parameters that need calibration,
which comes at the cost of introducing new uncertain parameters, and raises the question of models trustworthiness. This trade-
off between uncertainties and model complexity, highlighted in the field of ecological modeling by O'Neill and Rust (1979),
has been discussed recently by Saltelli (2019) who advises use of statistics to aid mathematical modeling via "*a systemic
appraisal of model uncertainties and parametric sensitivities*". There is, moreover, a growing appreciation of how sensitivity
analysis enhance understanding of the intricate physical processes involved in ice formation (Sheikholeslami et al., 2017). The
sensitivity of multiple-size-class frazil models to initial seeding, secondary nucleation and flocculation parameters has been
investigated by Wang and Doering (2005) and Smedsrud and Jenkins (2004), by simple modifications of specific parameters.
However, to the author's knowledge, probabilistic sensitivity analysis of frazil ice dynamics has never been performed. In

---

[1]SSC: Single Size Class

[2]MSC: Multiple Size Class





addition, comparison of (SSC) and (MSC) models in terms of uncertainty of output has never been tackled, despite that the prohibitive computational cost of the (MSC) model, making the (SSC) approach still relevant to many applications.

In this paper, we intend to bridge that gap by proposing an uncertainty and sensitivity analysis of single and multiple-class frazil ice models, using a of full description of parameters uncertainty (by probability density functions) and variance decomposition of model outputs. First, we introduce the models, and main hypotheses and discuss their implementation in section 2. The uncertainty assessment methodology is presented in section 3. We then rely on numerous field and experimental measurements to quantify the uncertainties of input parameters of the two frazil numerical models in section 4. Using Monte Carlo experiments, we next study propagation of these uncertainties on model outputs in section 5: in other words, the evolution of the frazil ice volume fraction and the water temperature. The evolution of output scatter is discussed and compared to the asymptotes of the dynamic systems. Finally, sensitivity analysis based on Sobol indices (Sobol, 2001) and aggregated Sobol indices enables us to propose a selection of the most influential parameters in both models. We conclude about this study's perspectives in section 6.

## 2 Frazil ice dynamics

This section introduces the continuum equations used to describe evolution of water temperature and frazil volume fraction, as well as their discrete counterparts and their numerical resolution under the assumption of well-mixed water bodies. The parameters of the models, that can be considered uncertain, are also introduced.

### 2.1 Mathematical description

Frazil crystals are supposed to be dics of radius $r$ and thickness $e$. The ratio between diameter and thickness, denoted $R = 2r/e$ is supposed to be constant as crystals grow. Let us define $n$ as the crystal number density, which corresponds to the number of crystals per unit volume per unit length in radial space. The total number of particles per unit volume is then defined as $N = \int_0^\infty n\,dr$. We also introduce the frazil volume fraction density as $c = nV$ in which $V = \pi r^2 e$ is the crystal volume. The volume proportion of the water-ice mixture occupied by frazil ice, i.e., the total volume fraction of frazil, is then defined as $C = \int_0^\infty c\,dr$. An incompressible water-ice mixture of velocity $\mathbf{u}$ and depth $h$ is considered. Following Daly (1984) the number density balance equation can be defined as

$$\frac{\partial n}{\partial t} + \mathbf{u}.\nabla n - \nabla.(\nu_c \nabla n) = \underbrace{-\frac{\partial}{\partial r}(Gn)}_{(1)} + \underbrace{\left(\dot{N}_T + \dot{N}_I\right)\delta(r - r_c)}_{(2)} - \underbrace{\frac{1}{V}\frac{\partial}{\partial r}(FVn)}_{(3)} - \underbrace{w_r \frac{\partial n}{\partial z}}_{(4)}, \tag{1}$$

in which $\nu_c$ is the frazil particles diffusivity, $w_r$ is the buoyancy velocity, and remaining right hand side terms are source terms described hereafter:

– The source term (1) represents the crystal size evolution due to thermal growth, resulting from the heat exchange between the supercooled water and ice particles. The crystals can be considered to grow mainly on their peripheral area, defined



by $a = 2\pi re$ (Daly, 1994). Frazil evolution is then driven by the radial growth rate $G$, defined as

$$\rho_i L_i G = \frac{Nu k_w}{\delta_T} \Delta T, \tag{2}$$

in which $\rho_i$ is the frazil ice density, $L_i$ is the latent heat of ice fusion and the right hand side is the heat exchange between the crystal of temperature $T_i$ and the surrounding water of temperature $T$. The latter is modeled as a function of the temperature delta $\Delta T = T_i - T$, the thermal conductivity of water $k_w$, the Nusselt number $Nu$, which represents the ratio between turbulent heat transfer and conduction heat transfer, and a thermal boundary layer length scale, denoted $\delta_T$. The crystal temperature $T_i$ is assumed to be equal to the freezing temperature denoted $T_f$. The Nusselt number can be described as a function of ratio $m^* = r/\eta$, where $\eta$ is the Kolmogorov length scale, which can be defined as a function of the turbulent dissipation rate $\varepsilon$ as $\eta = (\nu^3/\varepsilon)^{1/4}$, in which $\nu$ is the molecular viscosity of water (Daly, 1984). The Nusselt number formulation described by Holland et al. (2007) is used. For small particles, heat transfer is governed by diffusion and convection, and the Nusselt number can therefore be written as :

$$Nu = \begin{cases} 1 + 0.17 m^* P_r^{1/2} & \text{if} \quad m^* \le P_r^{-1/2} \\ 1 + 0.55 m^{*2/3} P_r^{1/3} & \text{if} \quad P_r^{-1/2} < m^* \le 1, \end{cases} \tag{3}$$

where $Pr$ is the Prandlt number, defined as the ratio between molecular and thermal diffusivity. For larger particles ($m^* > 1$), heat transfer is governed by turbulent mixing of the boundary layer around the crystal, and the Nusselt number is defined by

$$Nu = \begin{cases} 1.1 + 0.77 \alpha_T^{0.035} m^{*2/3} P_r^{1/3} & \text{if} \quad \alpha_T m^{*4/3} \le 1000 \\ 1.1 + 0.77 \alpha_T^{0.25} m^* P_r^{1/3} & \text{if} \quad \alpha_T m^{*4/3} > 1000, \end{cases} \tag{4}$$

where $\alpha_T$ is the turbulent intensity.

- Heterogeneous nucleation (growth from foreign nuclei) and secondary nucleation (birth of new nuclei due to breakage of parent crystals) are invoked to explain the continuous feed of frazil nuclei during the rapid intial frazil growth (Daly, 1984, 1994). Heterogeneous nucleation is mainly caused by impurities that come either from the water body itself (biological elements, suspended sediments, etc.) or by penetration of new nuclei from the atmosphere (meteorological conditions or artificial seeding during experiments). The source term (2) of Equation (1) models the introduction of new nuclei, as well as the secondary nucleation, where $\dot{N}_I$ is a seeding rate, and $\dot{N}_T$ the secondary nucleation rate function of collisions between crystals. $\delta(r - r_c)$ is the Dirac delta function and $r_c$ the critical radius (radius of new nuclei). Collisions are supposed to cause small nuclei to break off from parent crystals with a frequency of collision proportional to the crystal velocity relative to the fluid $U_r$ and the total number of particles $N$ that are contained in the volume swept by the crystal (Svensson and Omstedt, 1994). The secondary nucleation rate $\dot{N}_T$ is then defined as

$$\dot{N}_T = \tilde{n} \int_0^\infty \pi r^2 U_r n(r) dr, \tag{5}$$





where $U_r = \sqrt{4\varepsilon r^2/15\nu + w_r^2}$ is the geometric mean between turbulent velocity and buoyant rise velocity. $\tilde{n} = \max(N, n_{max})$ is the average number of particles per unit volume that take part in the collisions, and $n_{max}$ is a fitting parameter controlling the efficiency of the collision breeding. This parametrization was also followed by Smedsrud (2002), Smedsrud and Jenkins (2004), Wang and Doering (2005) and Holland and Feltham (2005).

- The source term (3) of Equation (1) is a flocculation source term supposed to represent the net effect of both flocculation and breakup, as introduced by Svensson and Omstedt (1994), who chose $F = F_0 r^2$, where $F_0$ is a constant. Their choice was motivated by the intuition that larger crystals are more prone to flocculate. However, as explained by Rees Jones and Wells (2018), this also depends on frazil concentration and hydrodynamic conditions, of which the effects on flocculation are still poorly characterized in literature. To be consistent with models in the literature, the formulation proposed by Svensson and Omstedt (1994) is chosen for this paper.

- The source term (4) of Equation (1) represents the buoyancy of frazil ice crystals, where $w_r$ is the rise velocity. In the present paper, it is set to $w_r = 32.8(2r)^{1.2}$, following Svensson and Omstedt (1994), who simplified the model proposed by Daly (1984). For consistency with previous modeling studies, we retain this simple formulation, even if many other formulations have been proposed as summarized by Morse and Richard (2009) and McFarlane et al. (2014).

Finally, the thermal balance of the water-ice mixture complements Equation (1). Supposing $C \ll 0$, the water fraction temperature equation is obtained :

$$\frac{\partial T}{\partial t} + \mathbf{u}.\nabla T - \nabla.(\nu_t \nabla T) = \frac{\phi}{\rho c_p} + \frac{\rho_i L_i}{\rho c_p} \int_0^\infty G a \, n \, dr, \tag{6}$$

where $T$ is the temperature of the water fraction of the water-ice mixture, $\nu_t$ the thermal diffusivity, $\rho$ the density of water, $c_p$ the specific heat of water and $\phi$ the net heat source resulting from heat exchanges with the atmosphere in W.m$^{-3}$.

In the following sections, a well-mixed water body is considered. Equations (1) and (6) are written in terms of space-averaged quantities. This assumption allows us to neglect the convection and diffusion operators and to focus on solving the average temperature and average frazil volume fraction. Therefore, the partial differential equations of frazil and temperature are simplified to a coupled set of Ordinary Differential Equations (ODEs) including only the source terms.

## 2.2 Radial space discretization

In this section we introduce a multiple-size-class (MSC) frazil ice model, which is a discrete version of Equations (1) and (6) in radial space. It consists of considering $m$ classes of constant radius chosen between a minimum and a maximum radius (Svensson and Omstedt, 1994). For each class $i$, the radius and the thickness are supposed to be equal to $r_i$ and $e_i = 2r_i/R$ ($1 \le i \le m$). The peripheral area as well as the surface and volume of frazil crystals are defined as $a_i = a(r_i) = 2\pi r_i e_i$, $s_i = s(r_i) = 2\pi(r_i e_i + r_i^2)$ and $V_i = V(r_i) = \pi r_i^2 e_i$, respectively. A log-Uniform discretization of the radial space is chosen as in previous studies (Svensson and Omstedt, 1994; Wang and Doering, 2005; Rees Jones and Wells, 2018). The number of crystals of class $i$ is noted $n_i$, and similarly, the volume fraction of crystals of class $i$ is noted $c_i$. The total volume fraction





of frazil, and total number of particles, are then defined as $C = \sum_{i=1}^{m} c_i$ and $N = \sum_{i=1}^{m} n_i$ with $c_i = n_i V_i$, respectively. The discrete version of Equation (1) written in terms of volume fraction balance reads:

$$\frac{dc_i}{dt} = \underbrace{V_i \left( \Gamma_{i-1} c_{i-1} + (\Lambda_i - \Gamma_i) c_i - \Lambda_{i+1} c_{i+1} \right)}_{(1)} + \underbrace{\tau_i}_{(2)} + \underbrace{\beta_{i-1} c_{i-1} - \beta_i c_i}_{(3)} - \underbrace{\gamma_i c_i}_{(4)} \quad (1 \leq i \leq m), \tag{7}$$

with the boundary conditions $V_0 = V_{m+1} = \Gamma_0 = \Gamma_m = \Lambda_{m+1} = \beta_0 = \beta_m = 0$. The discrete versions of the source terms introduced in Equation 1 are defined following previous work (Svensson and Omstedt, 1994; Wang and Doering, 2005; Holland and Feltham, 2005):

  – The thermal growth source term (1) is composed of thermal growth and melt functions $\Gamma_i$ and $\Lambda_i$, which are defined as:

$$\Gamma_i = H \frac{G_i a_i}{V_i \Delta V_i}, \tag{8}$$

$$\Lambda_i = (1 - H) \frac{G_i s_i}{V_i \Delta V_{i-1}}, \tag{9}$$

  where $G_i = G(r_i)$, $\Delta V_i = V_{i+1} - V_i$ and $H = He(T_f - T)$, with $He$ the Heaviside function allowing to a switch between melting or freezing. We suppose that frazil crystals grow from their peripheral area $a_i$ but melt from their surface $s_i$ (Holland and Feltham, 2005).

  – The crystal birth source term (2), composed of the secondary nucleation and seeding, reads

$$\tau_{i \neq 1} = -\zeta \alpha_i c_i, \tag{10}$$

$$\tau_{i=1} = \tau_s V_1 / h + \sum_{j=2}^{m} \alpha_j c_j, \tag{11}$$

  where $\zeta = V_1 / V_i$ is a coefficient to conserve crystal volume, $\alpha_i = \pi \tilde{n} U_r(r_i) r_i^2$, $\tilde{n} = \max(N, n_{max})$ and $\tau_s$ a constant seeding rate in $m^{-2}.s^{-1}$.

  – The flocculation source term (3) is defined as $\beta_i = a_f r_i / r_1$, where $a_f$ is a flocculation coefficient in $s^{-1}$.

  – The buoyancy of crystals is simplified into a gravitational removal sink term (4) defined as $\gamma_i = -w_r(r_i) a_d / h$, in which $h$ is the water depth. We introduce a coefficient $a_d$, to account for the uncertainty of the rise velocity and gravitational removal process.

Finally, writing the thermal growth source term as $S_i = V_i \left( \Gamma_{i-1} c_{i-1} + (\Lambda_i - \Gamma_i) c_i - \Lambda_{i+1} c_{i+1} \right)$, one can write the discrete version of Equation (6) as

$$\frac{dT}{dt} = \frac{\phi}{\rho c_p} + \frac{\rho_i L_i}{\rho c_p} \sum_{i=1}^{m} S_i, \tag{12}$$

A general discrete frazil ice model is implemented in the present work (Equations 7 and 12) so that one can easily retrieve the formulations developed in previous papers. For example, supposing $H = 1$, $a_d = 1$, $\tau_s = 0$ and writing $c_i = n_i V_i$, Equation (7) is equivalent to Equation (4) in Wang and Doering (2005) and Equation (1) in Svensson and Omstedt (1994) with $\zeta = 1$. By also neglecting flocculation ($a_f = 0$), it is equivalent to Equation (10) in Rees Jones and Wells (2018).



## 2.3  Single-size-class simplification

A simplified approach is to take a single-size-class (SSC) composed only of particles of radius $\overline{r}$ representative of the whole
crystal distribution. A simplified set of equations for the frazil volume fraction and water temperature can then be written as

$$\frac{dC}{dt} = GaN + \frac{1}{h}\left(V\tau_s - w_r a_d C\right), \tag{13}$$

$$\frac{dT}{dt} = \frac{\phi}{\rho c_p} + \frac{\rho_i L_i}{\rho c_p} GaN, \tag{14}$$

where $a = a(\overline{r})$, $V = V(\overline{r})$, $w_r = w_r(\overline{r})$ and $G = G(\overline{r})$.

## 2.4  Numerical methods to solve governing equations

Let us denote $t_k = t_0 + k\Delta t$ the time at iteration $k$, $t_0$ the initial time and $c^k = c(t_k)$. A semi-implicit theta scheme is imple-
mented for the MSC model, with a constant time step $\Delta t$. It consists of an Euler explicit time discretization of the left-hand
side derivative, while taking $c_i = \theta c_i^{k+1} + (1-\theta)c_i^k$ in the right-hand side. Choosing $\theta = 0$ leads to a fully explicit time scheme
while choosing $\theta = 1$ is equivalent to an implicit scheme on $c$. The non linear terms are treated semi-implicitly i.e. $G_i = G_i(t_k)$
in order to retrieve a linear system of the form $A[c_1^{k+1}, ..., c_m^{k+1}]^T = B$ in which $A$ and $B$ are two matrices defined in Appendix
A. The temperature balance equation is solved with a forward Euler time scheme. The semi-implicit time scheme is subject to
a stability condition function of the smallest radius. In the present study, we found that, for the range of radius tested in the
sensitivity analysis, decreasing the time step below $\Delta t = 0.25s$ did not impact the results, so a value of $0.25s$ was retained for
all simulations.

An important aspect of the numerical frazil ice models presented in this paper is the need to provide a non-zero initial
condition for the frazil volume fraction (in absence of seeding: i.e. $\tau_s = 0$). In the case of MSC models, Svensson and Omstedt
(1994) assumed a uniform initial number of particles $n_0$ in each radius class i.e. $n_i(t_0) = n_0$ $(1 \leq i \leq m)$. We followed the
same principle to initialize the system, but fixed the number of initial particles at zero for classes with a radius exceeding a
threshold $r_0$: i.e. $n_{r_i \leq r_0}(t_0) = n_0$. To be able to compare SSC and MSC models, we worked in terms of initial volume fraction
of frazil $C(t_0) = C_0$. The system was then initialized with $n_0 = C_0 / \sum_{r_i \leq r_0} V_i$ and $c_i(t_0) = n_i(t_0)V_i$ $(1 \leq i \leq m)$. As the
initial setup has a big influence on the results (Holland and Feltham, 2005), $C_0$ and $r_0$ are considered in the following sections
as uncertain parameters.

To test convergence in terms of the number of radius classes, we make it vary from $m = 10$ to $m = 1000$. The results,
presented in Figure 1, are consistent with observations by Rees Jones and Wells (2018) that the system requires $m \gtrsim 100$
to converge. Note that Svensson and Omstedt (1994) and Wang and Doering (2005) took $m = 20$ and $m = 40$, respectively.
It should also be noted that convergence in the number of classes depends on the initialization method of the system, as
highlighted by Holland and Feltham (2005). To perform all the numerical simulations required for a complete sensitivity
analysis, $m = 100$ was chosen as a trade-off between numerical convergence and computational cost.



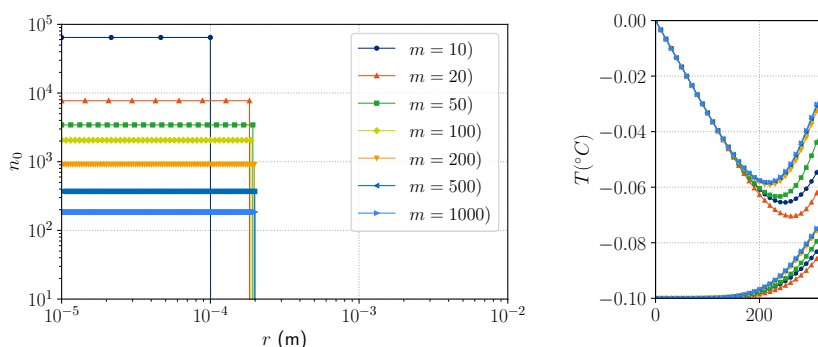

**Figure 1.** MSC model convergence in the number of classes with $C_0 = 4.5 \times 10^{-8}$, $r_0 = 0.2$mm, $r_{min} = 10^{-5}$m, $r_{max} = 10^{-3}$m, $dt = 0.1$s, $\phi = -1400$ W.m$^{-3}$, $\varepsilon = 1.2 \times 10^{-3}$ m$^2$.s$^{-3}$, $\alpha_T = 0.0876$, $n_{max} = 8 \times 10^6$ m$^{-1}$, $a_f = 10^{-4}$ s$^{-1}$ $\tau_s = 0$ m$^{-2}$.s$^{-1}$ and $a_d = 0$. Initial distribution of crystals (left) and water temperature and frazil volume fraction vs time (right).

## 2.5 Study cases and model parameters

In the present study, we focus on the evolution of water temperature and total frazil volume fraction in a supercooled, well mixed
water body of depth $h = 1$m. To focus on the frazil modeling rather than heat budget, uncertainties deriving from exchanges
with the atmosphere are neglected and the cooling rate $\phi$ is considered deterministic and constant over time in all experiments.
Nonetheless, different values of $\phi$ were tested, ranging from $-50$ to $-1000$W.m$^{-3}$, to test the variability of the results in
different cooling rate situations. As described in previous sections, frazil ice models are driven by many parameters, some of
which are subject to a significant degree of uncertainty. The list of parameters considered in the present study for conducting the
uncertainty analysis is shown in Table (1). Some physical properties are considered constant and are taken at $T = 0°C$ such that
$\nu = 1.792 \times 10^{-6}$ m$^2$.s$^{-1}$, $\rho = 999.82$ kg.m$^{-3}$, $\rho_i = 916.8$ kg.m$^{-3}$, $L_i = 3.35 \times 10^5$ J.kg$^{-1}$, $c_p = 4.1855 \times 10^3$ J.kg$^{-1}$.K$^{-1}$,
and $k_w = 0.561$ W.m$^{-1}$.K$^{-1}$.

Taking the SSC model as a reference, one can simply visualize the main feature of frazil dynamic systems. When time is
close to zero, the initial temperature decrease rate is defined by $\lim_{t \to 0} \dot{T} = \phi/\rho c_p$. In the absence of seeding and gravitational
removal (i.e. $\tau_s = 0$ and $a_d = 0$), the temperature decreases to the maximum supercooling point, characterized by a maximal
temperature depression, denoted $\theta = \max(T_f - T)$, and a critical time, denoted $t_c$. After the maximum supercooling, the frazil
production rate releases more heat compared to that released in the atmosphere, leading to an increase in water temperature
which tends toward freezing point. Finally, when time tends to infinity, there is a balance between the frazil growth source term
and the cooling rate. This leads to a linear increase in frazil volume fraction at a constant rate i.e. $\lim_{t \to +\infty} \dot{C} = -\phi/\rho_i L_i$.
This typical evolution of water temperature and frazil volume fraction is described in Figure (2) in which frazil volume fraction
asymptote, denoted $C^\infty$, is defined as :

$$C^\infty = C_0 - \phi t/\rho_i L_i. \tag{15}$$





**Table 1.** Description of uncertain parameters of the frazil ice models.

| Parameter | Unit | Description | Category | Model |
|:---:|:---:|:---:|:---:|:---:|
| $C_0$ | - | Initial frazil volume fraction | Initial condition | Both |
| $r_0$ | m | Initial maximum radius | Initial condition | MSC |
| $r_{min}$ | m | Minimum radius | Discretization | MSC |
| $r_{max}$ | m | Maximum radius | Discretization | MSC |
| $\overline{r}$ | m | Mean radius | Discretization | SSC |
| $R$ | - | Diameter to thickness ratio | Source term ① | Both |
| $\delta_T$ | m | Thermal growth length scale | Source term ① | Both |
| $\varepsilon$ | $\mathrm{m^2.s^{-3}}$ | Turbulent dissipation rate | Source terms ①, ② | Both |
| $\alpha_T$ | - | Turbulent intensity | Source term ① | Both |
| $n_{max}$ | $\mathrm{m^{-3}}$ | Secondary nucleation efficiency cap | Source term ② | MSC |
| $\tau_s$ | $\mathrm{m^{-2}.s^{-1}}$ | Seeding rate | Source term ② | Both |
| $a_f$ | $\mathrm{s^{-1}}$ | Flocculation coefficient | Source term ③ | MSC |
| $a_d$ | - | Buoyancy coefficient | Source term ④ | Both |

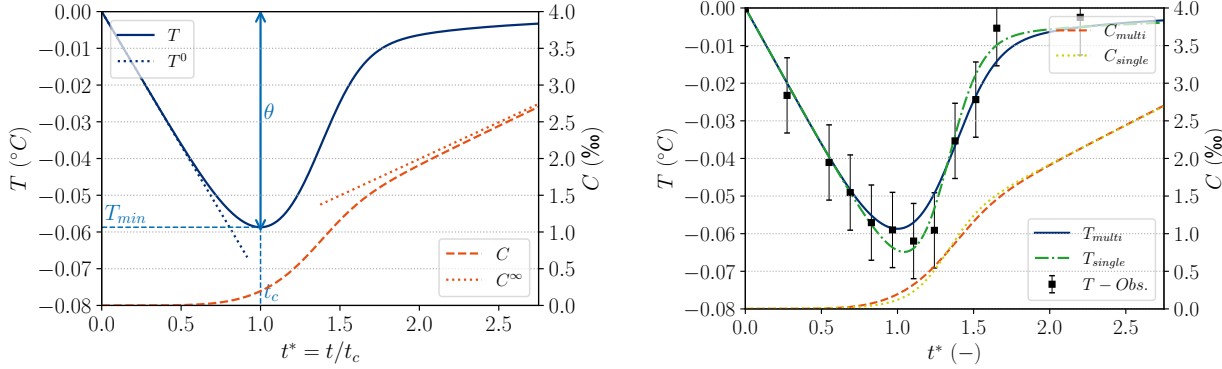

**Figure 2.** Typical evolution of water temperature and frazil volume fraction vs time (left) and reproduction of Carstens (1966) case I experiments (right) with $dt = 0.1$s, $\phi = -1400$ W.m$^{-3}$, $\varepsilon = 1.2 \times 10^{-3}$ m$^2$.s$^{-3}$, $\alpha_T = 0.0876$, $n_{max} = 8 \times 10^6$ m$^{-1}$, $a_f = 10^{-4}$ s$^{-1}$ $\tau_s = 0$ m$^{-2}$.s$^{-1}$, $a_d = 0$, $\overline{r} = 3.8 \times 10^{-4}$m, $R = 10$, $C_0 = 5.1 \times 10^{-6}$ (SSC) and $C_0 = 4.5 \times 10^{-8}$ (MSC), $r_0 = 0.2$mm, $r_{min} = 10^{-5}$m, $r_{max} = 10^{-3}$m, $m = 100$.

By introducing seeding and gravitational removal, i.e. $\tau_s \neq 0$ and $a_d \neq 0$, the frazil ice long term asymptotic is modified, and we have $\lim_{t \to +\infty} \dot{C} = -\phi/\rho_i L_i + (V\tau_s - w_r a_d C)/h$. Considering $\phi$ and $\tau_s$ as constant over time, this yields a convergence of the frazil volume fraction towards a finite limit, defined by the ratio between the buoyancy removal and the production rate

 

due to thermal growth and seeding, which reads :

$$C^\infty = \frac{h}{a_d w_r}\left(-\frac{\phi}{\rho_i L_i} + \frac{V\tau_s}{h}\right). \tag{16}$$

It should be observed that the steady states are not affected by the crystal growth rate. Similar observations were discussed by Rees Jones and Wells (2018) for the MSC model.

In the present study, we analysed the uncertainty of input parameters in the main transient and steady-state features of the frazil ice models with and without gravitational removal. We started the simulation with a water temperature equal to the freezing temperature (which is supposed to be equal to zero in the absence of salinity : i.e. $T_f = 0°C$). We checked that all the simulations ran beyond the critical time. For the range of parameters considered there, it was found that choosing a final time of $t_f \simeq 1$h was enough to capture the transient features of the ODE systems.

Note that both the SSC and MSC models are able to reproduce the experiments of Michel (1963), Carstens (1966) and (Clark and Doering, 2004) as shown by Svensson and Omstedt (1994) and Wang and Doering (2005). In Figure (2) we give an example of the results of both SSC and MSC models compared to Carstens (1966) experimental results (Case I).

## 3    Probabilistic framework for uncertainty analysis

The objective of this section is to describe the mathematical tools necessary to study the uncertainties of SSC and MSC frazil
ice models. An uncertainty study of a numerical model can be performed in three steps (Sudret, 2007). The first step consists in identifying the uncertain parameters, and characterizing the probabilities of occurrence for their values through Probability Density Functions (PDFs), which is referred to as *quantification of uncertainty sources*. The second step concerns the *propagation of uncertainties* through the interest models, generally using sampling techniques (e.g., Monte Carlo) to obtain possible values of the target outputs (here frazil ice concentration, water temperature) and compute their statistical estimates (mean,
standard deviation, percentiles, etc.). The third step, called *sensitivity analysis*, focuses on ranking the uncertain parameters in terms of influence on the target output. In the following, formulations for the statistical estimates and sensitivity analysis indices are given, with a summary of elements essential to understand this paper. Curious readers can find theoretical details in Soize (2017) and Sudret (2007). All uncertainty analysis computations were performed with the OpenTURNS library (version 1.18) developed by a partnership between Airbus Group, EDF R&D, IMACS, ONERA and Phimeca Engineering (Baudin
et al., 2016).

### 3.1    Uncertainty quantification

Let $\mathbf{X} = (X^1, \ldots, X^{n_X})$ be the vector of uncertain parameters of the frazil ice model, where $n_X$ is the number of uncertain parameters. Let $f$ be the interest frazil model, either SSC or MSC models presented in Equations (14) and (12) for temperature and (13) and (7) for frazil volume fraction. The output of the interest models are the frazil volume fraction and water tempera-
ture discrete time series i.e. $T(t_k)$ and $C(t_k)$ ($1 \le k \le n_t$), which we will generally denote $\mathbf{Y} = (Y^1, \ldots, Y^{n_t})$ below for the



sake of simplicity. Variables $\mathbf{X}$ and $\mathbf{Y}$ are linked through frazil models $f$ such that $\mathbf{Y} = f(\mathbf{X}, \mathbf{d})$, where $\mathbf{d}$ is a deterministic vector, i.e. fixed parameters in contrast with the uncertain set of inputs $\mathbf{X}$.

To undertake the uncertainty studies, both the inputs $\mathbf{X}$ and the interest outputs $\mathbf{Y}$ are considered to be random variables. This means, taking the example of $\mathbf{X}$, that they are characterized with probabilities of occurrence for given values $\mathbf{x}$ denoted $\mathbb{P}(\mathbf{X} = \mathbf{x})$. These probabilities link to the PDF denoted $p_{\mathbf{X}}$ through $\mathbb{P}(\mathbf{X} \in E \subseteq D_{\mathbf{X}}) = \int_E p_{\mathbf{X}}(\mathbf{x}) d\mathbf{x}$, where $E$ is a subset from the space of all possible values $D_{\mathbf{X}}$. Each element $X^i$ and $Y^k$ of the input and output vectors is hence characterized by a PDF.

The results of the uncertainty analysis are directly linked to the UQ study specification and consequently to the description of the uncertain input parameters $\mathbf{X}$. Thus, special attention is paid to propose a relevant quantification of uncertainty sources. This particular point is addressed in section (4), in which frazil literature is explored to provide adequate bounds and PDFs for each parameter identified in Table (1).

### 3.2 Uncertainty propagation of random variables

In the following, we suppose that PDFs of inputs $\mathbf{X}$ are known, in contrast to those of $\mathbf{Y}$. The objective is therefore to characterize the latter by estimating statistical indicators as the PDF's mean and standard deviation. To achieve this, a Monte Carlo sampling method can be used.

The Monte-Carlo technique consists in propagating a random generation of input variables, sampled with respect to their PDFs, through the model to estimate possible values for the output random vector $\mathbf{Y}$. The resulting sampling of the input random vector $\mathbf{X}$ is a matrix of size $N \times n_X$ where each row, denoted $\mathbf{x}_i = \{x^1, \ldots, x^{n_X}\}$, represents a possible configuration of the frazil model. The output realizations i.e. $\mathbf{y}_i = \{y_i^1, \ldots, y_i^{n_t}\}$, are generated by $N$ successive deterministic simulations with corresponding inputs. For a sampling of size $N$, the output of the model is a $N \times n_t$ matrix: $\mathbf{Y} = [y_i^k]_{i,k} = [f^k(\mathbf{x}_i, d)]_{i,k} \in \mathbb{R}^{N \times n_t}$. Each row of the matrix represents one output time series for a given set of input parameters, while each column represents the output realizations at a given time $t_k$.

Statistical estimators of the response can then be computed. The Monte-Carlo estimation of the mean, denoted $\widehat{\mu}_{Y^k}$ and standard deviation, denoted $\widehat{\sigma}_{Y^k}$ at time $t_k$ reads:

$$\widehat{\mu}_{Y^k} = \frac{1}{N} \sum_{j=1}^{N} y_j^k \quad \text{and} \quad \widehat{\sigma}_{Y^k} = \sqrt{\frac{1}{N} \sum_{j=1}^{N} \left(y_j^k - \widehat{m}_{Y^k}\right)^2} \tag{17}$$

These estimations converge to the true values following the law of large numbers (conditioned by the existence of corresponding PDF's first and second moment, i.e. expectation and variance (Soize, 2017)). The convergence order of the Monte-Carlo method is given by the central limit theorem leading to decrease of the confidence intervals' sizes proportional to $\sqrt{N}$. In the results of this paper, confidence intervals of the estimated mean and standard deviation are computed using a bootstrap method.

### 3.3 Sensitivity analysis using Sobol indices

Sensitivity analysis is essential in understanding numerical models (Razavi et al., 2021). It aims at quantifying the impact of inputs variables imprecision on the accuracy of the model output variables. Conventional approaches to Global Sensitivity





Analysis (GSA) imply the stochastic estimation of statistical moments and indices are classically achieved with the Monte-Carlo technique. For a given set of independent input parameters, the ANOVA (ANalyses Of VAriance) decomposition allows us to compute the variance of output $\mathbf{Y} = f(\mathbf{X}, \mathbf{d})$, for each time step $t_k$ $(1 \leq k \leq n_t)$ as:

$$Var[Y^k] = \sum_{i=1}^{n_X} V_i(Y^k) + \sum_{i<j} V_{ij}(Y^k) + \cdots + V_{1\ldots n_X}(Y^k), \tag{18}$$

where :

$$V_i(Y^k) = \text{Var}[\mathbb{E}[Y^k|X^i]]$$
$$V_{ij}(Y^k) = \text{Var}[\mathbb{E}[Y^k|X^i, X^j] - \mathbb{E}[Y^k|X^i] - \mathbb{E}[Y^k|X^j]]$$
$$= \text{Var}[\mathbb{E}[Y^k|X^i, X^j]] - V_i(Y^k) - V_j(Y^k)$$

in which $\mathbb{E}[Y^k|X^i]$ represents the conditional expectation of $Y^k$ with the condition that $X^i$ remains constant.

To evaluate the influence of each input parameter, the so-called Sobol indices are used (Sobol, 2001). The first and second order Sobol indices are defined as follows, for $k \in \{1, \ldots, n_t\}$ and $i \in \{1, \ldots, n_X\}$:

$$S_i^k = \frac{V_i(Y^k)}{\text{Var}[Y^k]}$$
$$S_{ij}^k = \frac{V_{ij}(Y^k)}{\text{Var}[Y^k]} \tag{19}$$

The first-order Sobol index $S_i^k$ indicates the part of output variance explained by a single parameter $X_i$ without interactions, whereas second-order indices $S_{ij}^k$ quantify the part of variance of the output explained by the interaction between two inputs $X_i$ and $X_j$. The number of second order indices is given by $\binom{n_X}{2} = n_X(n_X - 1)/2$. When the number of input parameters is too large, it may be difficult to estimate second-order indices. In that case, we only estimate first-order and total indices. Total indices quantify the part of variance of the output explained by an input and its interactions with all the other inputs parameters. Total Sobol indices are defined as follows, for $k \in \{1, \ldots, n_t\}$ and $i \in \{1, \ldots, n_X\}$:

$$ST_i^k = S_i^k + \sum_{i \neq j} S_{ij}^k + \sum_{i \neq j, l \neq i, j \leq l} S_{ijl}^k + \cdots = \frac{VT_i(Y^k)}{\text{Var}[Y^k]} = 1 - \frac{V_{-i}(Y^k)}{\text{Var}[Y^k]} \tag{20}$$

where $V_{-i}(Y^k) = \text{Var}[\mathbb{E}[Y^k|X^1, \ldots, X^{i-1}, X^{i+1}, \ldots, X^{n_x}]]$.

To compute Sobol indices, the method proposed by Saltelli (2002) is used, in which two independent samples of size $N$, denoted $\mathbf{A}$ and $\mathbf{B}$ are generated. Both can be written as matrices (cf. Equation 21) in which each line is a realization of the random vector $\mathbf{X}$. A third matrix, denoted $\mathbf{C}^i$ is then created by replacing only the column $i$ of the matrix $\mathbf{A}$ by the column $i$ of the matrix $\mathbf{B}$ (cf. Equation 21).

$$\mathbf{A} = \begin{pmatrix} x_1^{A,1} & x_2^{A,1} & \cdots & x_{n_X}^{A,1} \\ x_1^{A,2} & x_2^{A,2} & \cdots & x_{n_X}^{A,2} \\ \vdots & \vdots & \ddots & \vdots \\ x_1^{A,N} & x_2^{A,N} & \cdots & x_{n_X}^{A,N} \end{pmatrix}, \mathbf{C}^i = \begin{pmatrix} x_1^{A,1} & \cdots & x_i^{B,1} & \cdots & x_{n_X}^{A,1} \\ x_1^{A,2} & \cdots & x_i^{B,2} & \cdots & x_{n_X}^{A,2} \\ \vdots & & \vdots & \ddots & \vdots \\ x_1^{A,N} & \cdots & x_i^{B,N} & \cdots & x_{n_X}^{A,N} \end{pmatrix}. \tag{21}$$




First order and total Sobol indices are computed using estimations of $V_i(Y^k)$ and $V_{-i}(Y^k)$ computed using samples $\mathbf{A}$, $\mathbf{B}$ and $\mathbf{C}^i$ and noted $\hat{V}_i(Y^k)$ and $\hat{V}_{-i}(Y^k)$ respectively. These estimations are defined as follows:

$$
\hat{V}_i(Y^k) = \frac{1}{N-1} \sum_{j=1}^{N} \tilde{f}(\mathbf{B}_j)\tilde{f}(\mathbf{C}_j^i) - \left( \frac{1}{N} \sum_{j=1}^{N} \tilde{f}(\mathbf{A}_j) \right) \left( \frac{1}{N} \sum_{j=1}^{N} \tilde{f}(\mathbf{B}_j) \right),
$$

$$
\hat{V}_{-i}(Y^k) = \frac{1}{N-1} \sum_{j=1}^{N} \tilde{f}(\mathbf{A}_j)\tilde{f}(\mathbf{C}_j^i) - \left( \frac{1}{N} \sum_{j=1}^{N} \tilde{f}(\mathbf{A}_j) \right) \left( \frac{1}{N} \sum_{j=1}^{N} \tilde{f}(\mathbf{B}_j) \right),
$$

$$\tag{22}$$

where $\tilde{f}$ is the centered model defined by $\tilde{f} = f - \overline{f}$ in which $\overline{f}$ is the mean of the combined output samples $f(\mathbf{A})$ and $f(\mathbf{B})$. To compute the second-order Sobol indices, an additional matrix is used, noted $\mathbf{C'^j}$ which is created by replacing only the column $i$ of the matrix $\mathbf{B}$ by the column $i$ of the matrix $\mathbf{A}$. Then the estimation $\hat{V}_{ij}(Y^k)$ is computed as:

$$
\hat{V}_{ij}(Y^k) = \frac{1}{N-1} \sum_{m=1}^{N} \tilde{f}(\mathbf{C}_m^i)\tilde{f}(\mathbf{C}_m'^j) - \frac{1}{N} \sum_{m=1}^{N} \tilde{f}(\mathbf{A}_m)\tilde{f}(\mathbf{B}_m) - \hat{V}_i(Y^k) - \hat{V}_j(Y^k). \tag{23}
$$

For a sample size $N$, estimation of the first-order and total Sobol indices requires $(n_X + 2) \times N$ simulations.

For multivariate outputs, the indices can be aggregated as proposed by Gamboa et al. (2014). The aggregated first order and total Sobol indices are defined as:

$$
AS_i = \frac{\sum_{k=1}^{n_t} V_i(Y^k)}{\sum_{k=1}^{n_t} \mathrm{Var}[Y^k]}
$$

$$
AST_i = \frac{\sum_{k=1}^{n_t} VT_i(Y^k)}{\sum_{k=1}^{n_t} \mathrm{Var}[Y^k]}
\tag{24}
$$

This means that Sobol indices $S_i^k$ and $ST_i^k$ quantify the influence of $X_i$ on $Y$ and time $t_k$, while the aggregated indices $AS_i$ and $AST_i$ quantify the influence of $X_i$ over the whole time series of $Y$.

# 4 Uncertainty source quantification

Many laboratory experiments have been carried out to better understand frazil ice dynamics as summed up by Barrette (2020, 2021). These experiments, as well as field measurements, help us define parameter variability. In this section, we consider the uncertain parameters listed in Table (1), and deduce adequate variability and PDFs for uncertainty propagation and sensitivity analysis using the aforementioned data. The main geometrical properties of crystals and radial space discretization are first discussed, as well as initial concentration. We then review all the parameters involved in frazil source terms in the same chronology as presented in section (2).

## 4.1 Crystals' geometrical properties

Description of the frazil crystals' shape has been the subject of several field and laboratory measurements (Arakawa, 1954; Daly, 1984, 1994). The crystals have been described as thin discs that grow mainly from their peripheral area. More recently,




photos have brought valuable confirmation of the disc shape of frazil crystals, but also highlighted the complexity in larger flocs' geometry (Clark and Doering, 2006; Ghobrial et al., 2012; McFarlane et al., 2014, 2015; Schneck et al., 2019). Let us recall that, in both SSC and MSC models, discretization is done in radial space. Therefore, either the ratio $R$ or the thickness $e$ must be considered constant to fully describe disc-shaped particles. In the present study, frazil crystals are assumed to have the same aspect ratios, which is consistent with previous studies hypothesis (Svensson and Omstedt, 1994; Holland et al.,

2007). Note that Rees Jones and Wells (2018) considered constant thickness instead, but highlighted a weak dependency of the thermal growth to aspect ratio. Let us discuss the uncertainties associated with the choice of the radius discretization as well as the diameter to thickness ratio.

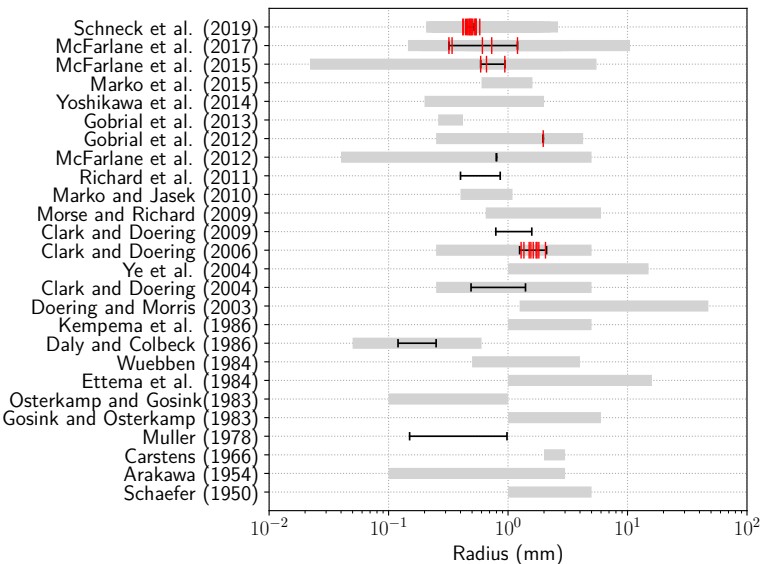

**Figure 3.** Particle size ranges (thick grey line), mean range (thin dark line) and log-normal distributions' mean radius (red vertical ticks) reported in field or laboratory experiments (Schaefer, 1950; Arakawa, 1954; Carstens, 1966; Gosink and Osterkamp, 1983; Osterkamp and Gosink, 1983b; Ettema et al., 1984; Wuebben, 1984; Kempema et al., 1986; Daly and Colbeck, 1986; Doering and Morris, 2003; Ye et al., 2004; Ye and Doering, 2004; Clark and Doering, 2004, 2006, 2009; Marko and Jasek, 2010; Ghobrial et al., 2012; McFarlane et al., 2012; Ghobrial et al., 2013; McFarlane et al., 2015; Kempema and Ettema, 2016; McFarlane et al., 2016, 2017; Schneck et al., 2019)

Either a mean radius or a radius space discretization must be specified for SSC and MSC models, respectively. In both cases, observed radius distribution and particle size, reported in a many studies, are taken into consideration. For the sake of

synthesis, we analyzed the corresponding publications from 1950 to 2019, and report the available observations in Figure 3. This figure is complementary to Table 7 of McFarlane et al. (2017), who summarize particle sizes from both laboratory and field measurements. Daly and Colbeck (1986) and Clark and Doering (2006) reported log-normal distributions of particles in laboratory experiments, with a mean radius ranging from 0.12 to 0.25 mm and 0.49 to 1.4 mm, respectively. This was later confirmed by Ghobrial et al. (2012) and McFarlane et al. (2015) in the University of Alberta Cold Room Facility, with mean





size ranging from 0.66 to 0.94 mm. Schneck et al. (2019) recently published similar results under different salinities, with mean radius ranging from 0.45 to 0.52 mm. Log-normal distributions were also observed in field studies, with the mean ranging from 0.59 to 0.94 mm in a report by McFarlane et al. (2017). Gathering together all reported mean radius ranges, an uncertainty interval of $1.2 \times 10^{-4}$ to $2.1 \times 10^{-3}$ m was chosen for the mean radius in the SSC model. When their full variation range is considered, frazil crystal sizes are known to follow a log-normal distribution as previously argued, but there is no evidence or

reason for the mean radius being log-normal as well. The number of field and laboratory observations is still too small to fit an empirical PDF on the mean radius with reasonable confidence. Hence, a log-uniform PDF is chosen with the previously described bounds. Using Log-uniform approximate distributions allows us to explore by means of evenly distributed values, parameters that vary over several orders of magnitude.

For the MSC model, both a minimum and a maximum radius are to be determined for the bounds of the radial discretization.

The minimum radius can be determined from Lal et al. (1969) survival theory and is about $4\,\mu\text{m}$ (Mercier, 1984). The maximum size of frazil particles is about 1 mm to 5 mm according to Clark and Doering (2004), but frazil flocs can be significantly larger as shown by Schneck et al. (2019), and can range up to several cm (McFarlane et al., 2017). Svensson and Omstedt (1994) worked with $[4\times10^{-6}, 3\times10^{-3}]$ m while Wang and Doering (2005) and Rees Jones and Wells (2018) chose $[7\times10^{-5}, 5\times10^{-3}]$ m and $[5\times10^{-5}, 5\times10^{-2}]$ m, respectively. In literature, $r_{min}$ is taken to be between $10^{-6}$ and $10^{-4}$, so we chose an intermediate

order of magnitude of $r_{min} = 10^{-5}$ m. The same choice was made for the maximum radius, leading to a value of $r_{max} = 10^{-3}$ m. Both parameters were then considered as uncertain parameters with log-uniform PDFs and were taken to be within $[10^{-6}, 10^{-4}]$ for $r_{min}$ and within $[10^{-3}, 10^{-1}]$ for $r_{max}$.

The aspect ratio ranged from 6 to 13 in the study by Daly and Colbeck (1986), while there were greater values, from 12.2 to 16.33, in Clark and Doering (2004, 2006). Arakawa (1954) reported an even wider aspect ratio range, from 5 to 100. More

recently, McFarlane et al. (2014) reported aspect ratios ranging from 11 to 71 with a mean of 37 and a standard deviation of 11. Considering all these observations, an uncertainty range of 5 to 100 was chosen for the diameter-to-thickness ratio. As not enough data to properly fit a PDF on aspect ratios was provided, a uniform PDF is chosen for the sensibility analysis (Maximum Entropy Principle Soize (2017)).

Weak dependency of the thickness to the radius was reported by McFarlane et al. (2014), who suggest assuming a increasing

aspect ratio as discs grow instead of a constant aspect ratio. Not also that, as frazil ice forms larger flocs, the disc shape hypothesis commonly accepted in models does not hold anymore. This could lead to erroneous estimations of the thermal growth rate of larger flocs. However, neither the variability of the aspect ratio in the crystals' distribution, nor that of the shape, are considered in the present study.

## 4.2 Initial concentration

As previously mentioned, either a non-zero initial volume fraction or a non-zero seeding rate is required to trigger thermal growth in the model. In experimental facilities, frazil ice nuclei are sometimes artificially introduced in water to initiate frazil ice growth (Muller, 1978; Tsang and Hanley, 1985) but the initial concentration is rarely addressed and depends on the method of initial seeding. As such, a large uncertainty domain is considered in this study for the initial volume fraction of frazil, ranging





from $10^{-8}$ to $10^{-4}$, to account for the lack of data on this parameter. Additionally, a log-uniform PDF is retained since the

parameter range variation vary over several orders of magnitude.

For the MSC model, it is necessary to choose how the initial volume fraction is distributed over size classes. One possibility
is to initialize the system with log-uniform distributions, similar to the one observed in laboratory experiments. However,
observations of nuclei close to $r_c$ in size are limited. Consequently, authors have preferred simpler initialization methods,
whereby a constant number of crystals is distributed equally over all classes (Svensson and Omstedt, 1994; Wang and Doering,
2005; Rees Jones and Wells, 2018). Holland et al. (2007) argued that distributing the initial volume fraction over a range of
radii, thus changing the initial number of particles per class, significantly impacts results. They found that filling only the
first size class, as in Hammar and Shen (1995) has less impact on results and should therefore be the preferred initialization
method. Given poor evidence of initial predominance of each class in nature, we decided to test both initialization methods i.e.
distributing the initial volume fraction over a range of small radii as presented in section (2.4) then feeding only the first class
as suggested by Holland et al. (2007). For the first method, ice nuclei are supposed to be initially spread between the minimum
radius $r_{min}$ and a threshold radius $r_0$, which we suppose can only be smaller than the mean radius (cf. Figure 3). Therefore,
the threshold $r_0$ is considered uncertain within the bounds $[1.2 \times 10^{-4}, 2.1 \times 10^{-3}]$. For the second method, we set $r_0 = r_{min}$,
so that only the first receives the initial concentration.

### 4.3 Thermal growth and turbulent parameters

As shown in Equation (2), thermal growth (1) is mainly affected by two uncertain parameters, which are the Nusselt number and
the thermal boundary layer thickness. In this section, we discuss the uncertainty of both parameters. Since the Nusselt number
is being modeled via turbulent parameters, namely the turbulent dissipation rate and the turbulent intensity, their uncertainty is
also addressed.

As discussed by Rees Jones and Wells (2018), the thermal boundary layer thickness $\delta_T$ is not constant around a crystal and in
recent studies there has been an inconsistent scaling of thermal growth. Svensson and Omstedt (1994) chose the length scale as
$\delta_T = e$ while others have chose $\delta_T = r$ (Smedsrud and Jenkins, 2004; Holland et al., 2007), leading to a serious underestimation
of the growth rate. Jones and Wells (2015) have shown that there is a logarithmic dependency of the thermal growth on the
aspect ratio that favors the $\delta_T = e$ scaling. To account for the variability of the thermal boundary layer thickness, $\delta_T$ should be
taken as an uncertain parameter. However, it should be mentioned that, with the variation range being $\delta_T \in [e, r]$, the ANOVA
methodology (section 3.3) cannot be applied, since the set of uncertain inputs, which contains $\delta_T$ and $e/r$, cannot be considered
independent anymore. Sophisticated methods such as the Rosenblatt transformation (Soize and Ghanem, 2004) can be applied
to transform the dependent parameter spaces into independent Gaussian inputs. This is however out of the scope of this paper.
To overcome this difficulty, the sensitivity analysis is conducted with the most appropriate scaling: i.e., $\delta_T = e$, for both the
SSC and MSC models. Nevertheless, we propose to investigate the impact of scaling by considering $\delta_T \in [\min(e), \max(r)]$ to
avoid modeling dependency. A log-uniform PDF is used and $\min(e)$ and $\max(r)$ are estimated from their full variation ranges.
A third experiment was also carried out with the $\delta_T = r$ scaling. The three results are then compared for the SSC model.





Two main turbulence parameters are considered: turbulent dissipation rate $\varepsilon$ and turbulent intensity $\alpha_T$, both impacting the Nusselt number. For rivers, the dissipation rate can be estimated using friction velocity $u_* = \sqrt{gR_hS}$ such that:

$$\varepsilon = \frac{u_*^3}{\kappa R_h}\left[ln\left(\frac{u_* R_h}{\nu}\right) - 1\right] \tag{25}$$

where $g$ is gravity, $R_h$ the hydraulic radius, $S$ the slope of the river, $\kappa$ the von Karman constant (generally taken as equal to 0.4), and $\nu$ the viscosity of water. Daly (1994) summarized the dissipation rate for several experiments (Michel, 1963; Carstens, 1966; Tsang and Hanley, 1985; Muller, 1978) using Equation (25) and found values ranging from $7 \times 10^{-5}$ to $0.4667$ m$^2$.s$^{-3}$. A similar method was described by McFarlane et al. (2015) to estimate the dissipation rate in rivers leading to values ranging from $4.2 \times 10^{-4}$ to $1.4968$ m$^2$.s$^{-3}$. Schneck et al. (2019) summarized dissipation rates observed in oceans, ranging from $10^{-9}$

to $10^{-2}$ m$^2$.s$^{-3}$ and from $10^{-10}$ to $10^{-3}$ m$^2$.s$^{-3}$ in polar regions. In the present study, we consider dissipation rates varying between $10^{-9}$ and $1.5$ m$^2$.s$^{-3}$, which cover most flows encountered in rivers and oceans. Similarly, we consider a wide range of flows from low turbulence to high turbulence intensity which include most of the work presented in Figure (3), leading to a turbulent intensity ranging from $1\%$ to $20\%$. Log-uniform PDFs are considered for both the dissipation rate and the turbulent intensity. Note that turbulence influences not only the rate of growth of frazil crystals but also the rate of secondary nucleation,

thus impacting the frazil size distribution.

## 4.4 Other source terms

Following the same order as in Equation (1), let us discuss uncertain parameters involved in frazil source terms other than thermal growth: secondary nucleation (2), flocculation (3) and gravitational removal (4). It should be mentioned that almost no direct observations of these processes have been reported in the literature. Therefore, the definition of the uncertainty intervals

is mainly based on expert knowledge and past numerical experiments in which parameters were determined by comparison to observed radius distributions.

–  The uncertainty interval of the seeding rate is set after Daly (1994) to $[3 \times 10^{-1}, 10^{-4}]$ m$^{-2}$.s$^{-1}$, and to simplify the uncertainty analysis, is considered constant over time. For the secondary nucleation, Svensson and Omstedt (1994) proposed setting a common value for $n_{max}$ that would allow focus on calibration of the initial seeding. They found that

a value of $n_{max} = 4 \times 10^6$, along with initial seeding of the order of magnitude of $n_0 = 10^4$, gave satisfactory results compared to the experimental results of Michel (1963) and Carstens (1966). Wang and Doering (2005) found that fitting a single $n_{max}$ value for all experiments was unsatisfactory. Instead, they proposed fitting both the initial seeding and $n_{max}$, leading to values ranging from $2 \times 10^4$ to $2 \times 10^5$ for Clark and Doering (2004) experiments, and from $2 \times 10^4$ to $2 \times 10^5$ for Carstens (1966) experiments. Smedsrud (2002) proposed a different value of $n_{max} = 10^3$, which was also

used by Smedsrud and Jenkins (2004) and Holland and Feltham (2005).

–  Similarly, the flocculation parameter $a_f$ was calibrated to a value of $10^{-4}$ s$^{-1}$ by Svensson and Omstedt (1994), who compared the results of their simulation to the expected size distribution spectra (other parameters in their model were set to $m = 20$, $\phi = -1000$W.m$^{-3}$, $\varepsilon = 10^{-3}$ m$^2$.s$^{-3}$ and $n_i(t_0) = 10^4$ ($1 \le i \le m$)). From all previous calibration efforts,





one could choose $n_{max}$ from $10^3$ to $10^7$ and $a_f = 10^{-4}$. But given the fact that secondary nucleation and flocculation are
still very poorly understood and modeled, we choose large uncertainty intervals for these parameters. We added an order
of magnitude for the bounds leading to $n_{max}$ ranging from $10^2$ to $10^8$ and we suppose that flocculation can, depending
on the conditions, be negligible, so that $a_f$ was varied from $10^{-8}$ to $10^{-3}$. $n_{max}$ and $a_f$ are considered independent from
hydrodynamic parameters, and Log-uniform PDFs were chosen for both.

– The gravitational removal is affected by both buoyant rise velocity of frazil particles, and deposition process once par-
ticles reach the surface of the water column. Several attempts to measure the frazil rise velocity were done (Osterkamp
and Gosink, 1983a; Wueben, 1984; McFarlane et al., 2014) and many formulations were proposed as summarized by
McFarlane et al. (2014). Significant scatter can be observed in the data as shown on Figure (10) of McFarlane et al.
(2014), with velocities ranging from approximately 0.7 to 16 mm/s for a radius of 1 mm. Models exhibit significant
differences as well (cf. figure 4). To take into account uncertainties inherent to the choice of the rise velocity model, a
buoyancy parameter $a_d$ is introduced. A rise velocity envelope, combining the results of all models, can be defined by
upper and lower bounds denoted $w_{max}(r, R)$ and $w_{min}(r, R)$ (cf. figure 4). The interval of $a_d$ is defined from the mean
of the gap between the simplified formulation used the present study for $w_r$ (Svensson and Omstedt, 1994) and upper
and lower bounds of the rise velocity envelope. To avoid the modeling of dependencies, a constant value is taken for
$a_d$, even if the envelope depends on the radius and on the diameter to thickness ratio. Finally, we propose $a_d \in [a^-, a^+]$
in which $a^+ = \text{mean}(w_{max}(r, R)/w_r(r))$ and $a^- = \text{mean}(w_{min}(r, R)/w_r(r))$. By considering $r \in [10^{-5}, 10^{-2}]$m and
$R \in [5, 100]$, the following interval is obtained: $a_d \in [0.086, 1.51]$. Finally, it should be noted that this uncertainty quan-
tification is rather imprecise since the rise velocity dispersion depends on the shape of the particles and flow conditions.
Therefore, our analysis could be refined by taking into account dependencies. Also note that low values of $a_d$ could also
be justified from the uncertainty of the deposition process, which is not modelled in the present study.

## 4.5 Summary of uncertain parameters

To conclude the uncertainty quantification, all the uncertain parameters, their bounds, and PDFs are summarized in Table (2).

## 5 Uncertainty propagation and sensitivity analysis

In this section, we present the different Monte Carlo simulation cases considered for the uncertainty analysis, as synthesized
in Table (3). Then we discuss the results of the uncertainty propagation and the sensitivity analysis obtained for both SSC and
MSC models.

### 5.1 Propagation cases

Monte Carlo simulations are first carried out without seeding and gravitational removal ($\tau_s = 0$ and $a_d = 0$) for both (SSC) and
MSC models, which correspond to cases (1) and (2), respectively. Seeding and gravitational removal processes are subsequently





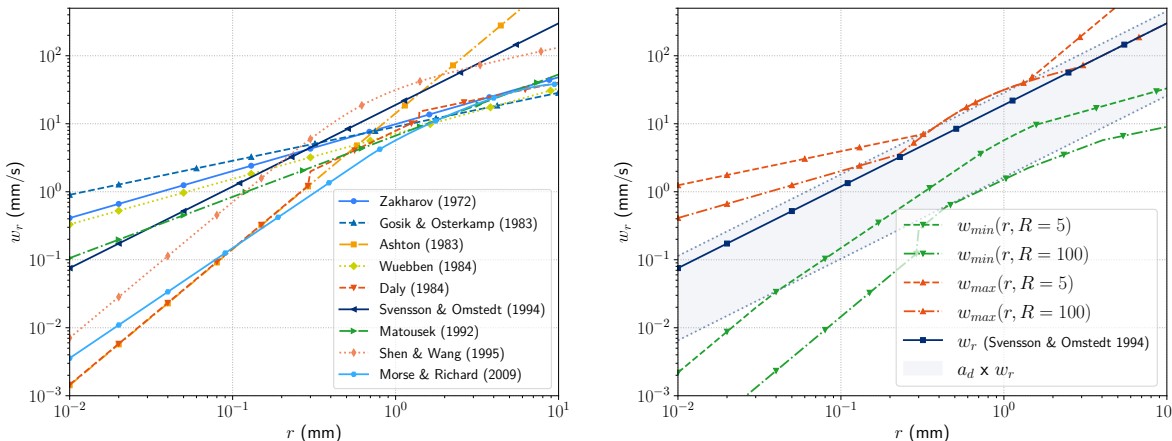

**Figure 4.** Comparison of frazil rise velocity models with $R = 10$ (left) and rise velocity envelope chosen for the uncertainty analysis (right) (Zacharov et al., 1972; Gosink and Osterkamp, 1983; Ashton, 1983; Wueben, 1984; Daly, 1984; Svensson and Omstedt, 1994; Matoušek, 1992; Shen and Wang, 1995; Morse and Richard, 2009)

**Table 2.** Uncertain parameters of the frazil ice models and their PDFs

| Parameter | Unit | Uncertainty interval | PDF |
|:---:|:---:|:---:|:---:|
| $C_0$ | - | $[10^{-8}, 10^{-4}]$ | Log-Uniform |
| $r_0$ | m | $[1.2 \times 10^{-4}, 2.1 \times 10^{-3}]$ | Log-Uniform |
| $r_{min}$ | m | $[10^{-6}, 10^{-4}]$ | Log-Uniform |
| $r_{max}$ | m | $[10^{-3}, 10^{-1}]$ | Log-Uniform |
| $\overline{r}$ | m | $[1.2 \times 10^{-4}, 2.1 \times 10^{-3}]$ | Log-Uniform |
| $R$ | - | $[5, 100]$ | Uniform |
| $\delta_T$ | m | $[7.34 \times 10^{-6}, 2.1 \times 10^{-3}]$ | Log-Uniform |
| $\varepsilon$ | $m^2.s^{-3}$ | $[10^{-9}, 1.5]$ | Log-Uniform |
| $\alpha_T$ | - | $[0.01, 0.2]$ | Log-Uniform |
| $n_{max}$ | $m^{-3}$ | $[10^2, 10^8]$ | Log-Uniform |
| $\tau_s$ | $m^{-2}.s^{-1}$ | $[3 \times 10^{-1}, 10^4]$ | Log-Uniform |
| $a_f$ | $s^{-1}$ | $[10^{-8}, 10^{-3}]$ | Log-Uniform |
| $a_d$ | - | $[0.086, 1.51]$ | Uniform |

considered in cases (3) and (4). For each case, the set of uncertain parameters is described in Table (2). When not in $X$,

parameters are considered deterministic with the following default values: $a_d = 0$, $\tau_s = 0$ $m^2.s^{-1}$, $\delta_T = e$, $r_{min} = 10^{-5}$ m and $r_{max} = 10^{-3}$ m.

Additional Monte Carlo simulations were carried out to examine the influence of specific parameters. For example, although $\delta_T = e$ is the preferred scaling and is chosen by default in all experiments, we also tested $\delta_T = r$ and $\delta_T \in [e, r]$ in experiments





**Table 3.** List of Monte Carlo simulations and their associated input random vectors $X$, sampling size, and number of function calls.

| Case | Model | Uncertain parameters | Case specificity | Sample size | No. of calls |
|------|-------|----------------------|------------------|-------------|--------------|
| 1  | SSC | $X = (C_0, \bar{r}, R, \varepsilon, \alpha_T)$ | - | $5 \times 10^4$ | $3.5 \times 10^5$ |
| 1b | SSC | $X = (C_0, \bar{r}, R, \varepsilon, \alpha_T)$ | $\delta_T = r$ | $5 \times 10^4$ | $3.5 \times 10^5$ |
| 1c | SSC | $X = (C_0, \bar{r}, R, \delta_T, \varepsilon, \alpha_T)$ | $\delta_T \in X$ | $5 \times 10^4$ | $4.8 \times 10^5$ |
| 2  | MSC | $X = (C_0, r_0, R, \varepsilon, \alpha_T, n_{max}, a_f)$ | - | $5 \times 10^4$ | $4.5 \times 10^5$ |
| 2b | MSC | $X = (C_0, r_0, r_{min}, r_{max}, R, \varepsilon, \alpha_T, n_{max}, a_f)$ | $r_{min}, r_{max} \in X$ | $6 \times 10^4$ | $6.6 \times 10^5$ |
| 2c | MSC | $X = (C_0, R, \varepsilon, \alpha_T, n_{max}, a_f)$ | $r_0 = r_{min}$ | $5 \times 10^4$ | $4 \times 10^5$ |
| 3  | SSC | $X = (C_0, \bar{r}, R, \varepsilon, \alpha_T, \tau_s, a_d)$ | $\tau_s, a_d \in X$ | $7 \times 10^4$ | $6.3 \times 10^5$ |
| 4  | MSC | $X = (C_0, r_0, R, \varepsilon, \alpha_T, n_{max}, \tau_s, a_f, a_d)$ | $\tau_s, a_d \in X$ | $6 \times 10^4$ | $6.6 \times 10^5$ |

(1b) and (1c) to investigate the impact on results. In experiments (2b) and (2c), modifications of the MSC model uncertain
parameters are also taken into account, in order to investigate the impact of the radius bounds $r_{min}$ and $r_{max}$, as well as the
alternate methods of initialization discussed in (4.2).

Statistical estimators are evaluated every 10s of physical time, leading to $n_t = 400$. To cope with the computational cost
of multi-class experiments we used clusters to run all simulations in parallel. The computation time of $4.5 \times 10^5$ multi-class
simulations with $m = 100$ is $\sim 24$ hours with 960 processes. A total of 4 million simulations were carried out for uncertainty
propagation.

## 5.2   Results without gravitational removal

By neglecting the seeding rate and gravitational removal source terms, the steady state corresponds to the constant frazil
production rate that only depends on the heat sink $\phi$, as shown in section (2.5). As expected, the statistical estimator time
series for temperature and frazil volume fraction, presented in Figure (5) for cases (1) and (2), show a very narrow scatter of
the output PDF at the start of simulation and at steady state. The results converge towards the two asymptotes ($T^0$ and $C^\infty$
respectively) of the mono-class ODE system: i.e., the constant cooling rate when $t \to 0$ (initial supercooling phase) and the
constant frazil production rate when $t \to \infty$ (recovery phase). However, for both SSC and MSC models, the transition between
the two asymptotes of the ODE system is spread out, and there is a significant difference in maximum supercooling between the
$5^{th}$, $25^{th}$, $75^{th}$ and $95^{th}$ percentiles. For the median, a maximum supercooling of $T(t_c) \simeq -0.018°C$ is reached at $t_c = 180s$
for the SSC model. The gap between the $5^{th}$ and $95^{th}$ percentiles maximum supercooling is $\Delta t_c = 14.8$ min and $\Delta\theta = 0.1°C$.
Note that the envelope obtained with the standard deviation gives a poor description of the output since its PDF is not normal.

Similar results were obtained with the MSC model, which recovers the same asymptote at steady state, however, there is a
slight residual scatter at recovery. For case (2), we observed slightly less scatter at the maximum supercooling than with the
SSC model, and the gap between the $5^{th}$ and $95^{th}$ percentiles maximum supercooling was $\Delta t_c = 13.7$ min and $\Delta\theta = 0.08°C$.
Time series of the first-order Sobol indices are presented in Figure (6) (see Appendix E and F for details). In Appendix C,
we also presente the first- and total-order Sobol indices at times $t_c$, $2 \times t_c$ and $t_f$ with $95\%$ confidence intervals, as well as



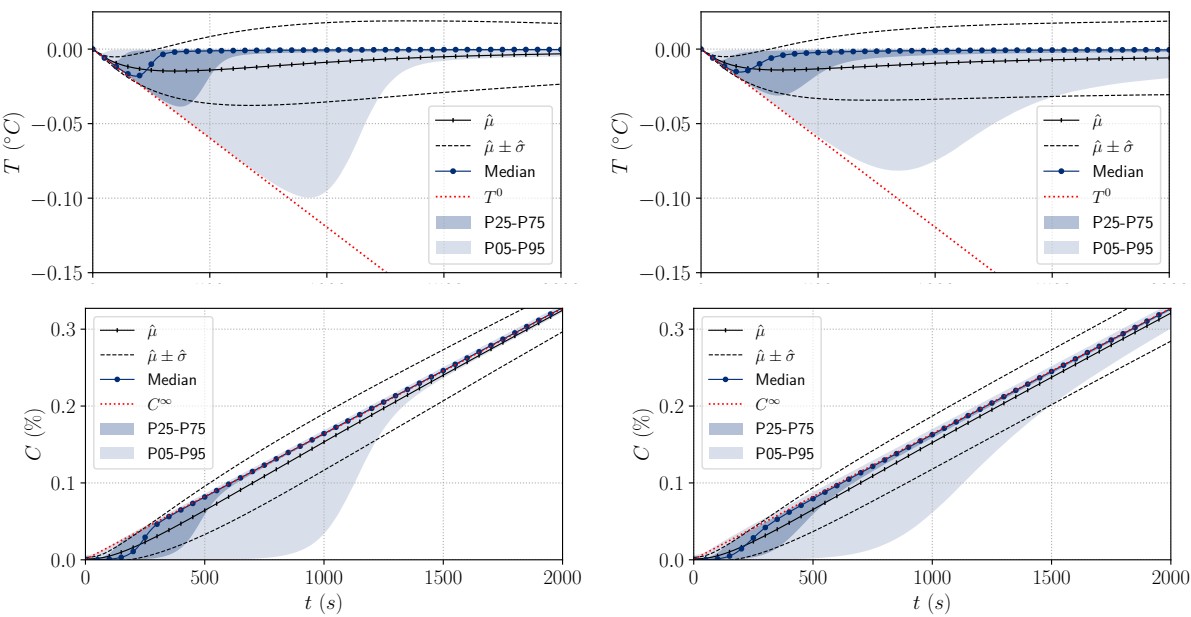

**Figure 5.** Uncertainty propagation results for SSC (case 1) on the left and MSC (case 2) on the right: mean, standard deviation, median, $5^{th}$, $25^{th}$, $75^{th}$ and $95^{th}$ percentiles are computed for $t_k (0 \leq k \leq n_t)$.

the aggregated Sobol indices, computed via Equation (24). The time evolution of Sobol indices of temperature and frazil were similar for both the SSC and MSC with the exception of the initial concentration, which, as expected had more impact on frazil volume fraction at the start of simulation.

For the SSC model, the initial concentration plays an important role at the start of the simulation. However, its influence quickly decreases, and the most influential parameter becomes the radius, with a peak influence reached at the time of maximum supercooling. Thus, using the average radius as the calibration parameter of the SSC model seems to be the most relevant choice. The most influential parameters after radius are diameter-to-thickness ratio and dissipation rate of turbulent kinetic energy. In both experiments and nature, turbulent dissipation rate is often better known than the initial concentration or crystals

shape. Therefore, one could then choose diameter-to-thickness ratio or initial frazil volume fraction as a secondary calibration parameters.

For the MSC model, the most influential parameters prior to maximum supercooling are initial concentration and maximum initial radius. Both are used to define the initial condition of the system. Using the initial distribution parameter to calibrate the transient phase until maximum supercooling seems to be the right approach. A first step in this endeavor, would be to specify

accurate values for initial distribution by comparison with what can be observed in nature, although this would require further research. At the recovery, the parameters of secondary nucleation and flocculation processes, both impacting steady-state crystal distribution, become more influential. However, the hierarchy of parameter is less obvious than for the (SSC) model.



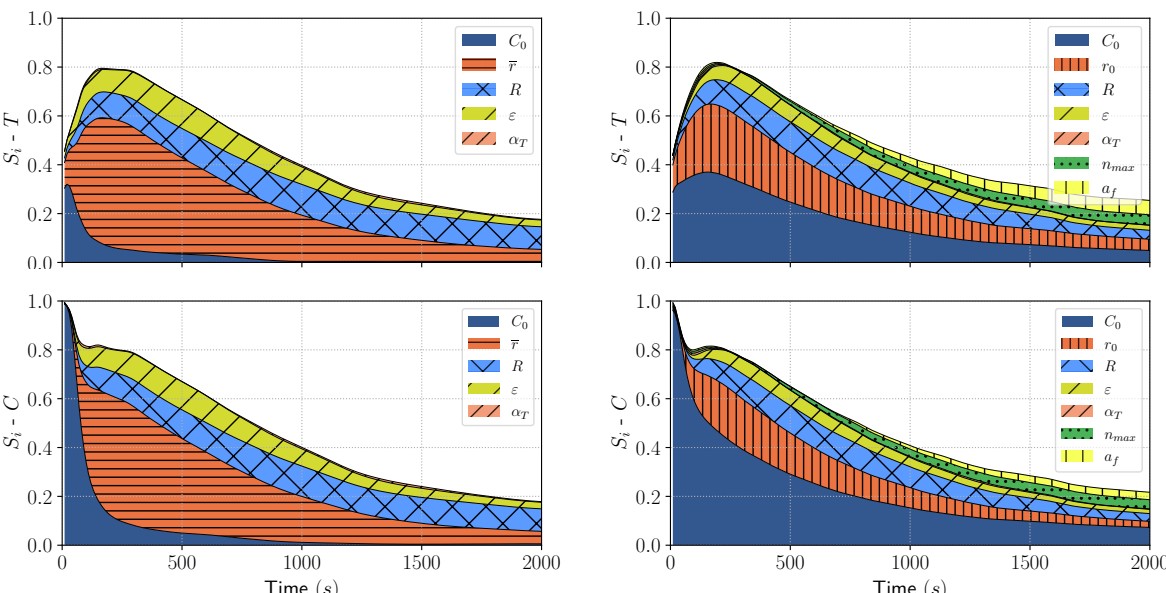

**Figure 6.** Time series of first-order Sobol indices ($S_i$) for temperature ($T$) and total frazil volume fraction ($C$) for SSC (case 1) on the left and MSC (case 2) on the right.

In addition, we observed strong interactions between parameters when the model reaches steady state, which might lead to difficulties in the calibration process. Aggregated Sobol indices summarized in Figure (7), confirm the relative influence of each parameter over the whole duration of the simulation, taking into account both the transient phase and steady state.

While the approach adopted by Svensson and Omstedt (1994): i.e., tweaking the values of $n_{max}$ and $a_f$, was adequate to calibrate the frazil distribution in (MSC) models, the present sensitivity analysis shows that it might not be the best option to calibrate water temperature and frazil total volume fraction. Results suggests that more focus should be on the initial condition to calibrate supercooling, by modifying initial seeding like it was done by Wang and Doering (2005) or by modifying the initial distribution itself. We therefore suggest calibrating the supercooling curve with the help of the initial distribution, along with secondary nucleation and flocculation parameters to calibrate the evolution of size distribution over time. Hopefully, recent observations of the transient evolution of frazil size distribution (McFarlane et al., 2015; Schneck et al., 2019), will provide the necessary data to carry out an optimal calibration of the identified parameters.

## 5.3 Influence of gravitational removal and seeding

Long-term evolution that does not take account of gravitational removal leads to infinite increase in frazil concentration as long as the cooling rate remains constant (cf. Equation 15). This asymptotic behavior of the models has never been observed in experiment nor in nature. Clark and Doering (2006) observed a peak in the number of particles per image they recorded, located shortly after maximum supercooling, after which there was a small decrease in the number of particles and a stagnation

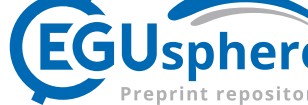

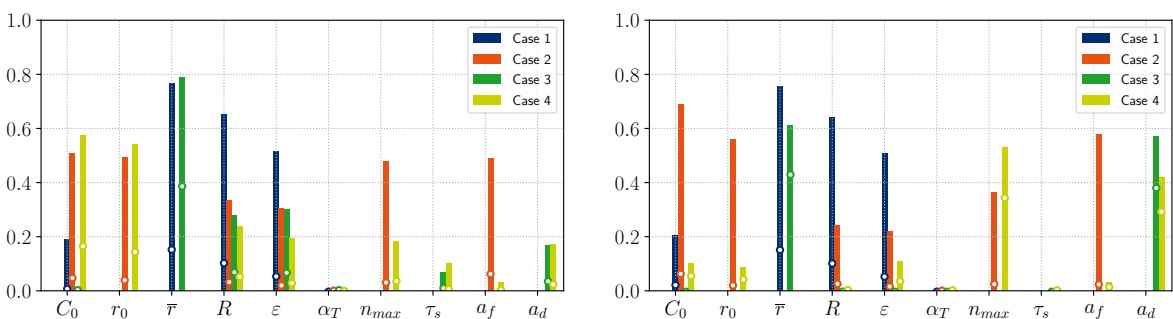

**Figure 7.** Aggregated first order Sobol indices (dots) and aggregated total Sobol indices (bars) for temperature ($T$) and total frazil volume fraction ($C$) for cases (1), (2), (3) and (4).

at residual supercooling. Similar observations were also reported by McFarlane et al. (2015) and Schneck et al. (2019). The

models in which only thermal growth is considered do not incorporate the required physics to properly reproduce what is observed. However by introducing gravitational removal, as shown in section (2.5), the models converge towards a constant frazil volume fraction (cf. Equation 16). In this section we analyze the results of the uncertainty propagation and sensitivity analysis for cases (3) and (4) which include the seeding rate and gravitational removal source terms.

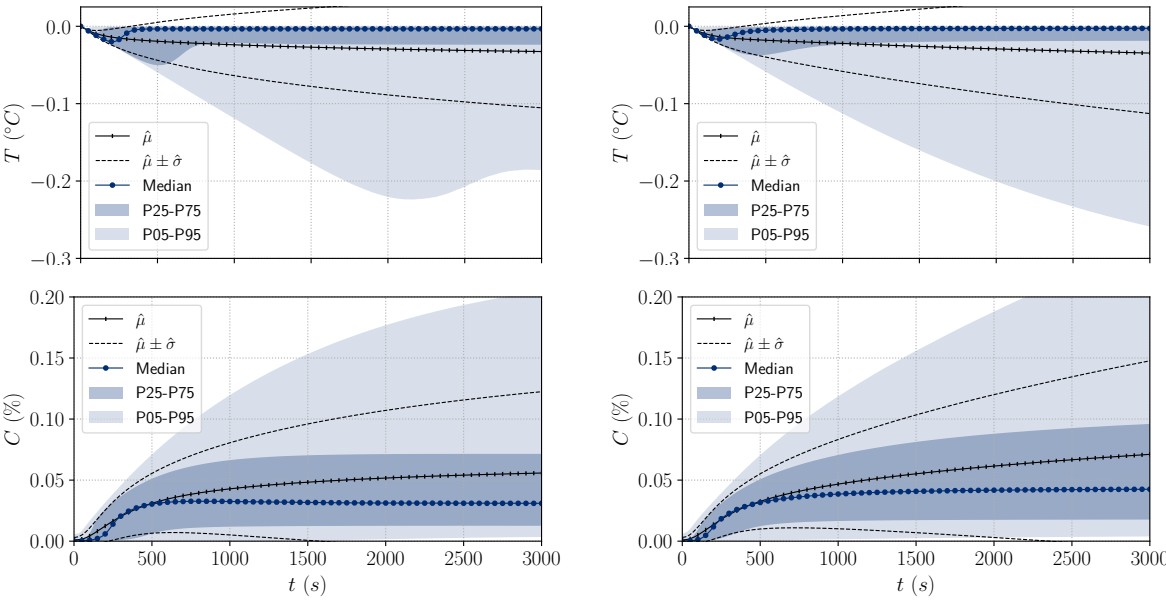

**Figure 8.** Uncertainty propagation results for SSC (case 3) on the left and MSC (case 4) on the right: mean, standard deviation, median, $5^{th}$, $25^{th}$, $75^{th}$ and $95^{th}$ percentiles computed for $t_k(0 \leq k \leq n_t)$.





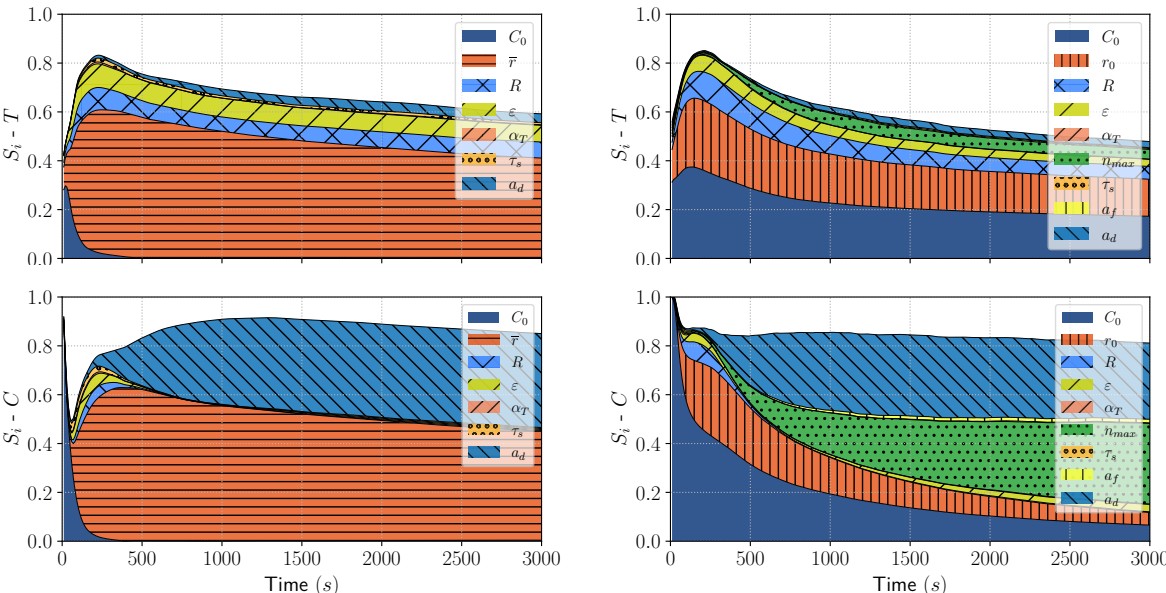

**Figure 9.** Time series of first-order Sobol indices ($S_i$) for temperature ($T$) and total frazil volume fraction ($C$) for SSC (case 3) on the left and MSC (case 4) on the right.

At steady state, the introduction of gravitational removal leads to wide scatter of both temperature and frazil as shown in
Figure (8) (see Appendix G and H for details). This is caused by the variation of the steady-state frazil volume fraction, which depends on the heat flux $\phi$ and model parameters inherent in secondary nucleation, flocculation and rise velocity. No significant difference is observed in the output scatter between SSC and MSC models similarly to cases (1) and (2). However, the median frazil volume fraction at steady state is slightly different with the two models.

With the SSC model at recovery, the first-order Sobol indices in Figure (9), show a major influence of the radius and
the buoyancy coefficient ($S_i$-$C = 0.381$ and $0.382$, respectively at $t = t_f$), the main parameters influencing the gravitational removal (cf. Equation 15). For the MSC model, the most influential parameters at recovery are $n_{max}$ and $a_d$ ($S_i$-$C = 0.37$ and $0.26$, respectively at $t = t_f$), which is coherent with Equation (27) and (29) of Rees Jones and Wells (2018). The hierarchy of the most influential parameters is similar to cases (1) and (2) prior to maximum suprcooling. However accurate modeling of the buoyancy velocity of frazil crystals is essential, as it has a very important influence on the long-term evolution of the
system, and therefore merits particular attention.

## 5.4 Maximum supercooling scatter

The results discussed above were obtained with a cooling rate of $-500$ W.m$^{-3}$. Several cooling rates, ranging from $-50$ to $-1000$ W.m$^{-3}$, were tested with case (1) to assess variations in maximum supercooling predictions. We found that the higher the cooling rate, the greater the scatter of the predicted maximum supercooling temperature, as presented in Figures (10) and





(11). This is the opposite for the time until maximum supercooling peak, where the higher the cooling rate, the lower the scatter in supercooling time. Note that the gap between $5^{th}$ and $95^{th}$ percentile maximum supercooling predictions is as much as $\Delta t_c = 42$ min with $\phi = -50$ W.m$^{-3}$, and $\Delta\theta = 0.142°C$ with $\phi = -1000$ W.m$^{-3}$ as shown in Figure (11).

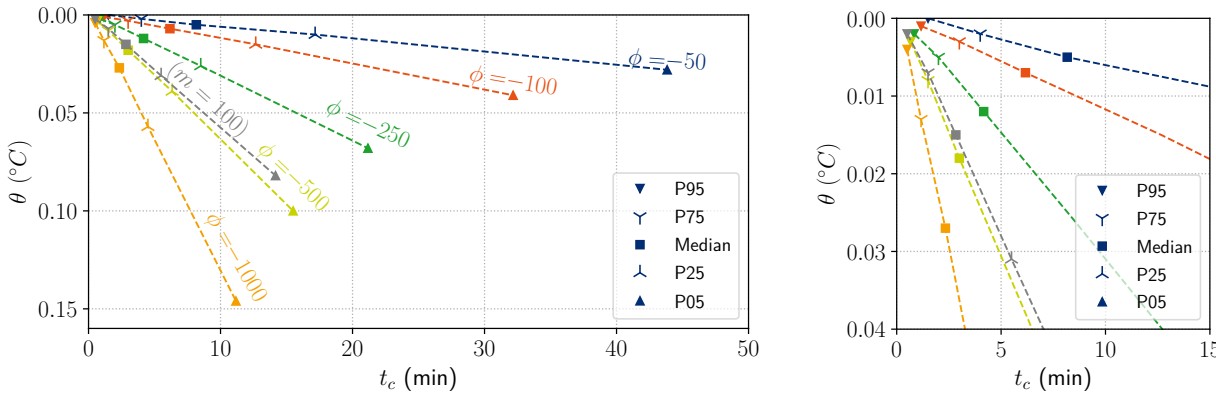

**Figure 10.** Maximum supercooling point scatter computed from median, $5^{th}$, $25^{th}$, $75^{th}$ and $95^{th}$ percentile time series at different cooling rates ($\phi = -50, -100, -250, -500,$ and $-1000$ W.m$^{-3}$) for case (1) and with $\phi = -500$ W.m$^{-3}$ for case (2).

  Results obtained from different scalings of the thermal boundary layer are also shown in Figure (11). A significant increase in scatter is observed for the $\delta_T = r$ compared to $\delta_T = e$ scaling, consistent with the fact that thermal growth is clearly under-
estimated (Rees Jones and Wells, 2018). With $\delta_T$ taken as an uncertain parameter with a log-unifrom PDF within the bounds $[\min(e), \min(r)]$, the result is also more widely scattered by the same order of magnitude as with $\delta_T = r$. The results highlight the significant influence of the choice of scaling for the boundary layer. Choice of scaling also explains inconsistencies in calibrated parameters in literature.

  Finally, let us discuss the results obtained with the MSC model, which need to be viewed from the standpoint of the the way
it is initialized. By taking the radius bounds as uncertain parameters (case 2b), we observe greater scatter between the $5^{th}$ and $95^{th}$ percentiles, and $\Delta t_c = 15.5$ min and $\Delta\theta = 0.09°C$ (cf. Figure 11). The constant feed of first-class nuclei due to secondary nucleation, in addition to the volume growth rate being higher for small classes, makes the initial distribution of concentration a determining choice. This also explains why the minimum radius has more influence on the results than the maximum radius as shown in Appendix D. The more initial concentration is attributed to the smallest classes, the quicker the model reaches steady
state, and the less scatter is observed in maximum supercooling time. The extreme case is when the initial concentration is only applied to the first class (case 2c), when an astonishingly narrow scatter of results is observed. In fact, the transient evolution is so quick that the model almost instantaneously converges towards its steady state. The median maximum supercooling is only $-0.001°C$ and is reached in only 20 seconds. This confirms the sensitivity test carried out by Holland and Feltham (2005), who suggested distributing initial concentration on one class. However, the results show that this type of initialization might
not be the best option, as it almost totally does away with transient evolution of the model.




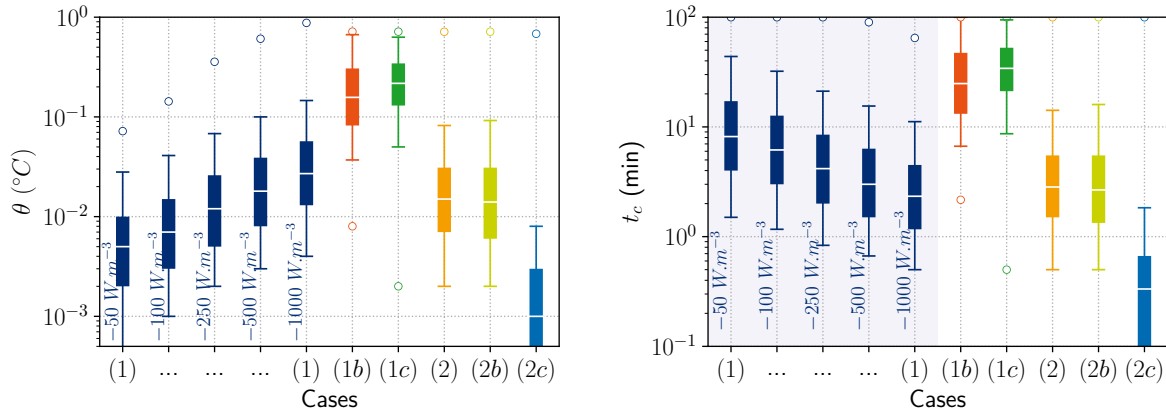

**Figure 11.** Maximum supercooling point scatter computed from median, $5^{th}$, $25^{th}$, $75^{th}$ and $95^{th}$ percentile time series at different cooling rates ($-50$, $-100$, $-250$, $-500$, and $-1000$ W.m$^{-3}$) for case (1), comparison between the choice of the length scale $\delta_T$ (cases 1b and 1c) and comparison between different initial conditions (cases 2b and 2c).

## 5.5 Limits and perspectives

The importance of the availability of qualitative field and laboratory data, to support the uncertainty quantification of model parameters, cannot be stressed enough. As we have shown, many model parameter uncertainties could not be characterized by means of direct data. We had to use expert knowledge as well as past numerical implementation (Daly, 1984, 1994; Svensson

and Omstedt, 1994; Smedsrud, 2002; Smedsrud and Jenkins, 2004; Wang and Doering, 2005; Holland and Feltham, 2005; Rees Jones and Wells, 2018). In part, this is because most physical processes, such as secondary nucleation or flocculation, are not directly measured, but instead are inferred from frazil distribution observations. Moreover, some parameters, such as the initial condition, are dependent on the discretization method. Consequently, several parameter uncertainty bounds and PDFs should be refined. Note that some parameters, such as $\delta_T$ and $a_d$, depend on other parameters (e.g., $r$ and $R$), adding a degree

of complexity to the sensitivity analysis. Innovative methods to tackle parameter dependency could move these barriers.

In this paper, we considered a well-mixed water body and a simplified gravitational removal sink term. Spatial variations for temperature and frazil may result in different conclusions. A poorly mixed water body, where the cooling rate at higher layers is more severe than at the bottom, would cause an heterogeneity in the initial formation of frazil. Additionally, the meteorological seeding of frazil nuclei occurs mainly at the free surface, increasing the heterogeneity. Furthermore, as the rise velocity is

higher for larger particles, the rise of frazil crystals and flocs yet again increases the heterogeneity of frazil on the vertical axis, altering the steady state distributions, as it was shown in the study by Hammar and Shen (1991). All these complex processes make the conclusions for a well-mixed body difficult to extrapolate to multidimensional cases. Further research therefore may be warranted into uncertainty quantification and sensitivity analyses of frazil ice models in non-well mixed conditions with multidimensional models. The emergence of efficient APIs in numerical tools such as TELEMAC-MASCARET (Goeury





et al., 2022), together with meta-modeling techniques that synthesize the essence of the multi-dimensional fields (Mouradi et al., 2021), could greatly assist such an undertaking.

Viewed from a different perspective, one may use the same probabilistic framework to compare modeling approaches for each process. This could be done by focusing on the volume fraction of each class, to obtain a good picture of how frazil is distributed over radius. In the same way that we considered time series of total frazil volume fraction as output, one could

easily transpose the analysis to multi-dimensional class volume fraction as output. Uncertainty propagation could then allow for the characterization of PDFs associated with each class, and sensitivity analysis would shed light on the properties of each model and how they affect frazil distributions. This could be a valuable tool for inferring new laws of secondary nucleation or flocculation by comparison to the observed evolution of frazil distributions.

## 6   Summary and conclusions

In this paper, two mathematical models for predicting of the evolution of frazil ice volume fraction and temperature have been studied. We developed a multiple-size-class MSC model, relying on the radial space discretization (Svensson and Omstedt, 1994) of frazil ice dynamic equations introduced by Daly (1984), which includes processes such as thermal growth, secondary nucleation, flocculation, seeding and gravitational removal. A simplified single-size-class SSC model, including only thermal growth, seeding and gravitational removal, was also developed for comparison. Properties of the two models, such as their

steady states, were highlighted, and details provided for their numerical resolution. We found that, for proper resolution of the transient phase with the MSC model, a class number of about 100 is necessary for convergence of the results, which corroborates the observations by Rees Jones and Wells (2018). Uncertainties in both models were then studied within a probabilistic framework. Aided by recent experimental and field studies of frazil ice, and also by numerical studies carried out in the recent years, the uncertainty of the main parameters of the frazil ice models were quantified. Various Monte Carlo experiments were

considered to propagate the uncertainties. Time dependent statistical estimates of the models' outputs (temperature and frazil volume fraction) were then analysed and a sensitivity analysis was carried out by means of a variance decomposition method.

Given the uncertainty bounds defined in the present study, SSC and MSC models yield very similar results for the prediction of water temperature and total frazil volume fraction. In the absence of gravitational removal, we have shown that the uncertainties have a great impact on the maximum supercooling and recovery time, but scarcely any impact on the steady state,

which is governed only by cooling rate. The more detailed physics of the multi-class model, although providing valuable new information on size distribution of the crystals, does not make it possible to obtain a more reliable estimate of water temperature and total frazil volume fraction in the transient phase. The development of MSC models raises the possibility that uncertainty may be removed from choosing a mean radius. We have shown, however, that scatter is similar somehow in both models, and derives from new uncertain parameters inherent in radial space discretization. Note that, in many numerical tools, modeling

frazil distribution requires the resolution of multiple advection-diffusion equations. Given the number of classes required for a model convergence, one can easily grasp the high numerical cost of using the MSC model for large scale, multi-dimensional applications. This makes the SSC model a relevant candidate for multi-dimensional frazil ice modeling, and the present study





shows that it is still a very good compromise between uncertainty and model complexity. However, it should be noted that such uncertainty in the MSC model could be overcome in future by a better estimation of the initial crystal size distribution.

The sensitivity analysis, allowed us to address with confidence the choice of calibration parameters. Relying on first- and total-order Sobol indices, we quantified the relative influence of each uncertain parameter on the output distribution, for both SSC and MSC models, and proposed a selection of parameters to be used for calibration. For the SSC model, the most influential factor for both temperature and frazil is the mean radius. Initial concentration played a secondary role although it was initially identified as a predominant factor. We therefore suggest using the average radius as the main calibration parameter.

The turbulent dissipation rate also plays a major role and as such should be specified with care. As it is often appropriately quantified, we suggest using initial concentration and diameter-to-thickness ratio as secondary calibration parameters. With the MSC model, we showed that the dispersion is somehow similar to what we observed for the SSC model but originates from new uncertain parameters. Thus, the most influential parameters on the transient phase are the parameters specific to the initial condition. However, once the steady state was reached, we observed an increasing influence of the secondary nucleation

and flocculation parameters. The long-term evolution of the system also showed increasing interactions between parameters, which can be explained by the balance in the physical processes involved in class interactions. When gravitational removal was introduced in the models, the stationary state was modified and the concentration converged towards a finite limit instead of diverging. In the case of the SSC model, the asymptotic limit is a function of the ratio between the gravitational removal term and the heat flux while in the case of the MSC model, the stationary state is also a function of the steady state radius

distribution (which depends on the balance between secondary nucleation, flocculation and gravitational removal). Our study, confirming previous asymptotic analyses, showed that both secondary nucleation and gravitational removal parameters are the most influential on total frazil volume fraction. The buoyant rise velocity, the uncertainty of which was rarely taken into account in previous modeling studies, should therefore be one of the main foci of future efforts to calibrate frazil ice models. Contrary to frazil, water temperature was mostly influenced by the initial condition, even at steady state. This should impel us to use

both water temperature and frazil volume fraction measurements to calibrate the models. Fortunately, recent laboratory and field studies (Schneck et al., 2019; McFarlane et al., 2015, 2017), particularly on the evolution of frazil distributions over time, offer precious data that can assist in developing appropriate calibration of the models. In this regard, using optimal calibration techniques, that allow consideration of both modeling and data uncertainties, would be a natural and complementary extension of the present study.





## Appendix A: Semi-implicit theta scheme matrices

In this section, the matrices $A$ and $B$ resulting from the semi-implicit time discretization of Equations 7 and 12 proposed in section 2.4 are provided. The matrix system reads: $A[c_1^{k+1},...,c_m^{k+1}]^T = B \Leftrightarrow$

$$
\begin{pmatrix}
a_{11} & a_{12} & a_{13} & & \cdots & & a_{1m} \\
a_{21} & a_{22} & a_{23} & 0 & \cdots & & 0 \\
0 & \ddots & \ddots & \ddots & & & \\
& & a_{ii-1} & a_{ii} & a_{ii+1} & & \vdots \\
\vdots & & & \ddots & \ddots & \ddots & 0 \\
& & & & & & a_{m-1m} \\
0 & & \cdots & & 0 & a_{mm-1} & a_{mm}
\end{pmatrix}
\begin{pmatrix}
c_1^{k+1} \\
\vdots \\
c_i^{k+1} \\
\vdots \\
c_m^{k+1}
\end{pmatrix}
=
\begin{pmatrix}
b_1 \\
\vdots \\
b_i \\
\vdots \\
b_N
\end{pmatrix},
\tag{A1}
$$

in which the diagonal terms are defined as:

$$a_{11} = 1 - \theta\Delta t\left(V_1(\Lambda_1 - \Gamma_1) - \beta_1 - \gamma_1\right),$$

$$a_{ii} = 1 - \theta\Delta t\left(V_i(\Lambda_i - \Gamma_i) - \beta_i - \gamma_i - \zeta\alpha_i\right) \quad \text{for} \quad 2 \le i \le m-1,$$

$$a_{mm} = 1 - \theta\Delta t\left(V_m\Lambda_m - \gamma_m - \zeta\alpha_m\right),$$

the lower off-diagonal terms are defined as:

$$a_{ii-1} = -\theta\Delta t\left(V_i\Gamma_{i-1} + \beta_{i-1}\right) \quad \text{for} \quad 2 \le i \le m,$$

the upper off-diagonal terms are defined as:

$$a_{1i} = -\theta\Delta t\alpha_i \quad \text{for} \quad 2 \le i \le m,$$

$$a_{ii+1} = \theta\Delta t V_i\Lambda_{i+1} \quad \text{for} \quad 2 \le i \le m-1,$$

and the matrix $B$ is defined as:

$$b_1 = c_1^k + \Delta t\tau_s V_1/h + (1-\theta)\Delta t\left[V_1\left((\Lambda_1 - \Gamma_1)c_1^k - \Lambda_2 c_2^k\right) - \beta_1 c_1^k - \gamma_1 c_1^k\right] + (1-\theta)\Delta t\sum_{j=2}^m \alpha_j c_j^k,$$

$$b_i = c_i^k + (1-\theta)\Delta t\left[V_i\left(\Gamma_{i-1}c_{i-1}^k + (\Lambda_i - \Gamma_i)c_i^k - \Lambda_{i+1}c_{i+1}^k\right) + \beta_{i-1}c_{i-1}^k - \beta_i c_i^k - \gamma_i c_i^k - \zeta\alpha_i c_i^k\right]$$
$$\text{for} \quad 2 \le i \le m-1,$$

$$b_m = c_m^k + (1-\theta)\Delta t\left[V_m\left(\Gamma_{m-1}c_{m-1}^k + \Lambda_m c_m^k\right) + \beta_{m-1}c_{m-1}^k - \gamma_m c_m^k - \zeta\alpha_m c_m^k\right],$$





## Appendix B:  Sturdiness of statistical estimators

680   Convergence of statistical estimators is addressed by running several Monte Carlo simulations with increasing sampling size. An example of the convergence of mean and standard deviation at different times is given in Figure B.1 for case (1). Similarly, testing of Sobol indices convergence is shown in Figure B.2 for case (1). Note that the 5th and 95th confidence intervals are systematically computed by a bootstrap method, and plotted in all Sobol index Figures.

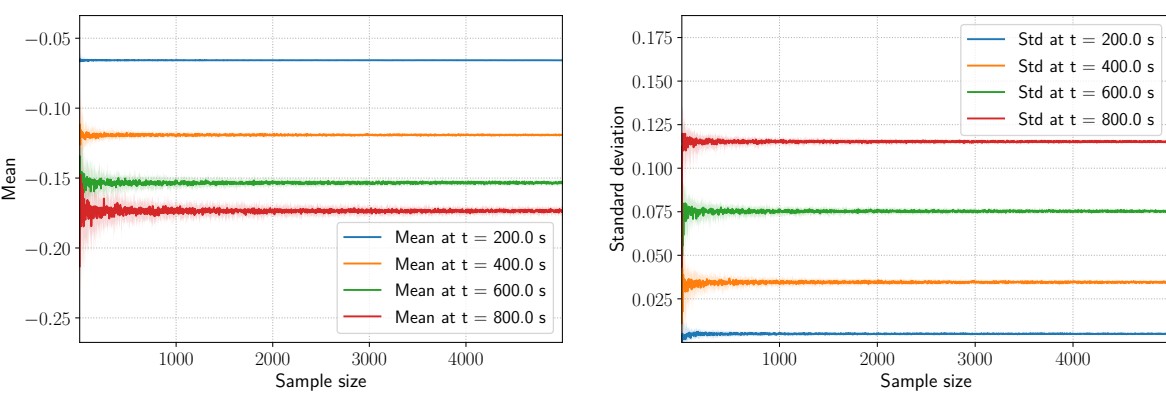

**Figure B.1.** Mean and standard deviation convergence for case (1) for temperature (left) and frazil (right).

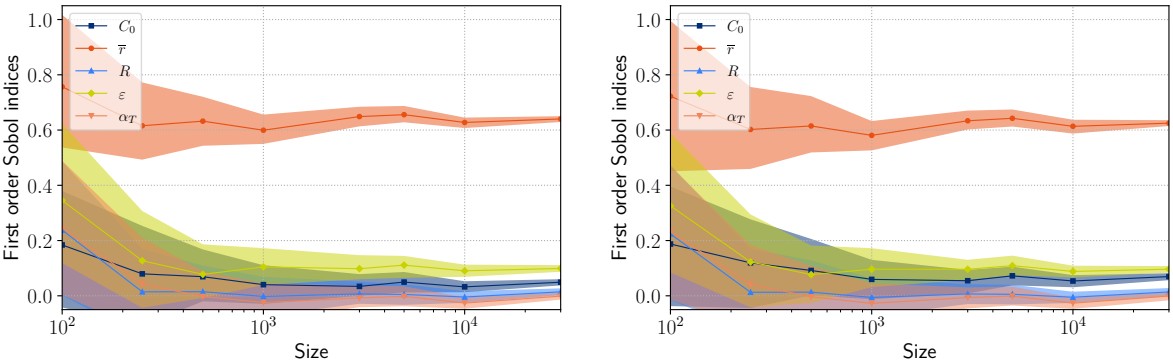

**Figure B.2.** First order Sobol indices convergence for case (1) for temperature (left) and frazil (right).



## Appendix C: Sobol indices and aggregated Sobol indices

**Figure C1.** First order and total order Sobol indices at times $t_{min}$, $2t_{min}$ and $t_f$ and aggregated Sobol indices for temperature (left) and frazil (right) the case (1), (2), (3) and (4) from top to bottom.





685 **Appendix D: Aggregated Sobol indices**

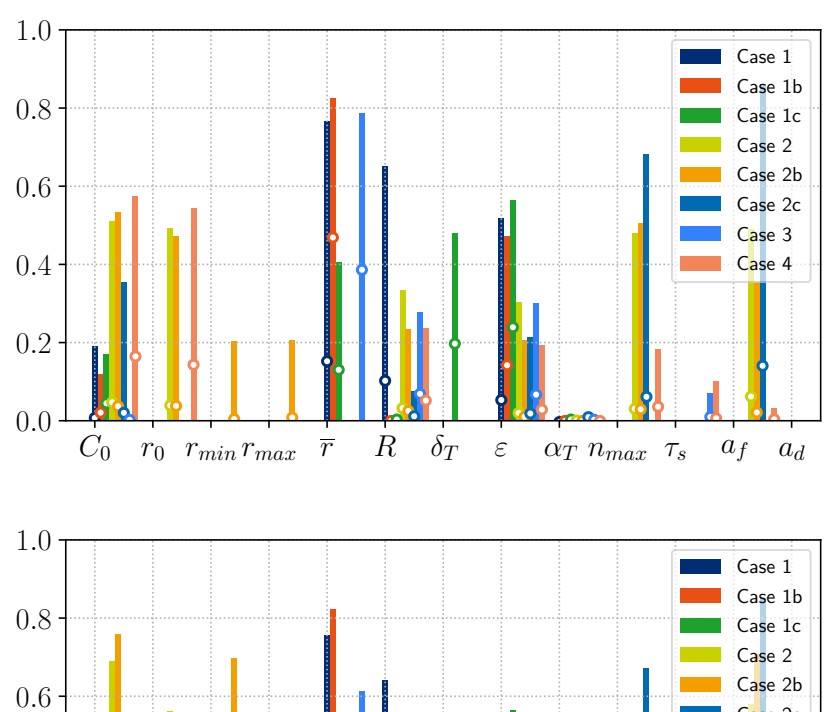

**Figure D1.** Aggregated first order Sobol indices (dots) and aggregated total Sobol indices (bars) for temperature ($T$) and total frazil volume fraction ($C$) for the case (1), (1b), (1c), (2), (2b), (2c), (3) and (4).





## Appendix E: Results of case (1)

**Figure E1.** Results of case (1). From top to bottom: uncertainty propagation result, first-order Sobol indices, first order Sobol indices with 95% confidence intervals, total order Sobol indices with 95% confidence intervals.



## Appendix F: Results of case (2)



**Figure F1.** Results of case (2). From top to bottom: uncertainty propagation result, first-order Sobol indices, first order Sobol indices with 95% confidence intervals, total order Sobol indices with 95% confidence intervals.





## Appendix G: Results of case (3)

**Figure G1.** Results of case (3). From top to bottom: uncertainty propagation result, first-order Sobol indices, first order Sobol indices with 95% confidence intervals, total order Sobol indices with 95% confidence intervals.





## Appendix H: Results of case (4)

**Figure H1.** Results of case (4). From top to bottom: uncertainty propagation result, first-order Sobol indices, first order Sobol indices with 95% confidence intervals, total order Sobol indices with 95% confidence intervals.





690 *Author contributions.* FS, CG, and RSM conceptualized the study. FS conducted the analyses, interpreted the results, and wrote the manuscript with input from CG and RSM.

*Competing interests.* The contact author has declared that none of the authors has any competing interests.

*Acknowledgements.* This research was funded by EDF. The authors gratefully acknowledge contributions from the OpenTURNS open-source community (Open source initiative for the Treatment of Uncertainties, Risks'N Statistics).





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
