# Peer review of "Uncertainty analysis of single- and multiple-size-class frazil ice models"

_EGUsphere, 2022_

## Referee Comment (RC1)

**Review of egusphere-2022-1162 by Mark Loewen**

**Uncertainty analysis of single- and multiple-size-class frazil ice models**

**Fabien Souillé, Cédric Goeury, and Rem-Sophia Mouradi**

I am referring to line numbers below.

Figure: In all figures with multiple plots you should label the plots (a), (b) etc. to avoid confusion.

74: Change "supposed to be" to "assumed to be".

191-193: Explain the reasoning for fixing the number of initial particles at zero for classes exceeding a radius threshold.

262: Define UQ.

Sec. 3.3: I recommend this be moved to an appendix.

384: I agree the uncertainty in the initial volume fraction is very large.

500: In Fig. 5 time is plotted in seconds so referring to minutes in the text is inconsistent. Also please check the value of 0.1 C I think it is inaccurate.

511: Are you referring to the median value of the time of maximum supercooling here?

514: I think for experiments this argument is valid but I am not convinced this is true in rivers. There are very few reliable measurements of dissipation in rivers available and the uncertainty could easily be an order of magnitude or larger.

518: Here you refer to "the initial distribution parameter" and it would be helpful to state also that these are $C_0$ and $r_0$.

520: In Table 1 $C_0$ and $r_0$ are categorized as "Initial conditions" and here they are referred to as initial distribution. I find this confusing.

521: You write "At the recovery" and later refer to a "recovery time" and these are both too vague. You define the "recovery phase" previously but this includes all time after steady state is reached. So clear terminology and clear definitions are required. In the same line you write "the parameters of secondary nucleation and flocculation processes". Please list all of these parameters here.

523: How did you observe interactions between parameters?

Figure 8: Supercooling of -0.2 C is quite extreme so some comments on these values are required. In the right plot please also comment on the fact that for P95 the maximum supercooling is not reached even after 3000 s.

546: List the parameters please.

552; Replace "coherent" with consistent?

554-555: Excellent point I agree.

559: Spelling changed to predicted.

560-562: This is a very limited discussion of Figure 10. Seems too brief - presumably there is more to discuss.

571-573: Awkward wording, please rewrite this sentence.

Figure 11; Add info to the caption to explain the symbols i.e., dots, error bars etc.

582: I think you mean quantitative here not qualitative.

591-599: Excellent discussion here.

599: Define API.

632: Replace relevant with suitable or promising?

640: I found it very interesting that you found that the turbulent dissipation rate plays a major role. Laboratory studies have found that the mean particle size varied with dissipation rate but the results are inconsistent. Can you use your model to examine this?

640-641: I do not agree that the dissipation rate is often appropriately quantified. Reliable measurements of dissipation rates in rivers are virtually non-existent and even in the lab it has not been accurately measured very often.

645-646: You write "The long-term evolution of the system also showed increasing interactions between parameters, which can be explained by the balance in the physical processes involved in class interactions". This was not clear to me since I do not think you explained this well in the paper.

652-653: Your conclusion regarding the rise velocity is a very significant result – well done!

---

## Referee Comment (RC2)

**Overall comments**

The manuscript analyses sources of uncertainty in modelling frazil ice. Frazil-ice processes are complex but are geophysically significant in a range of cryospheric settings. The manuscript falls within the scope of *The Cryosphere*. The manuscript analyses a wide body of literature in a thorough and careful fashion and includes lots of helpful review/analysis. It then undertakes a systematic formal uncertainty analysis which is a worthwhile novel advance in the field; in contrast previous analysis was more ad hoc and based on a limited exploration of parameter space. The manuscript compares single and multiple-size-class models and finds that (within the types of experiments considered), single-class models can be tuned to match the results of multiple-size-class models. Different sources of uncertainty in different types of models are systematically quantified and discussed. While the type of calculations are limited to examples motivated by simple laboratory experiments (and the picture may be more complex in the field), this study achieves significant and worthwhile results within its context. *The paper is well written, thorough and insightful and should be published subject to the minor comments below.*

**Minor comments**

- Code/data: I was not sure whether the code developed/used in this study was already available and if so where? The submission guidelines suggest:

  *Authors are encouraged to deposit software, algorithms, and model code in FAIR-aligned repositories/archives whenever possible. These research outputs are then cited in the manuscript using the received DOI and included in the reference list. The manuscript must then include a section entitled "Code availability" or, in the case of data and code, "Code and data availability".*

- L18: Could mention geophysical contexts like plumes of Ice Shelf Water under floating ice sheets.
- L48: Sentence probably needs splitting.
- L50: Perhaps add or substitute a geophysical example (e.g. from seismology)
- L56: A different kind of thing, but probabilistic methods are sometimes used in processing observational data (Frazer, E. K., Langhorne, P. J., Leonard, G. H., Robinson, N. J., & Schumayer, D. (2020). Observations of the size distribution of frazil ice in an Ice Shelf Water plume. *Geophysical Research Letters*, 47, https://doi.org/10.1029/2020GL090498)
- L74: The choice of $e$ to denote thickness was a bit confusing as the letter is usually reserved for Euler's number. I'm not sure if there is a particular precedent or motivation. Perhaps $e$ related to edge length? Similarly, the definition of $a$ above equation (2) might be somewhat confusing (as it denotes only part of the surface area, not the whole). If there is a source of both of these choices come from, it would be good to mention it.
- L92: not very clear what delta_T refers to at this stage (I see you come back and discuss it later, so it would be sensible to add some cross references and/or consider consolidating the discussion)

- L108-109: It seems to be assumed that seeding and secondary nucleation produce crystals of the same size. This might be as good as any other assumption but could be spelled out more clearly
- L118: while relying on this precedent is fine, I think the main issue is that the flocculation rate in this form is independent of the number of the number of crystals. The nucleation term written (2) in equation (1) will be proportional to the square of the number of crystals while the flocculation term written (3) will be linear in the number of crystals. [I would think that both these terms arise from the same type of processes and should both be quadratic.]
- L180-188: the reference to Appendix A could have been put nearer the start of the paragraph? The whole of this paragraph was relatively technical and could have been move the appendix, perhaps. It would have been good to have a brief explanation of how this scheme was chosen (especially because other studies didn't necessarily use the same method).
- Figure 1: I found the graphs quite cluttered and hard to read. Could experiment with different colour schemes, larger figure panels (there was quite a bit of white space), perhaps one/two less data series. Perhaps the left panel would have been more useful plotted at some later time instead (e..g t=300s).
- Table 1: I spotted n_max here but missed where it was discussed in the main text (I saw it in L160 but not discussed in paragraph starting L105). It is quite a significant fudge factor so needs discussing somewhere in words when the notation is first introduced.
- Section 3: I felt this could be moved to an appendix/supplement. It wasn't clear that anything was particular to this manuscript. Perhaps a one paragraph summary could go at the start of what is currently section 4.
- L360: Presumably the minimum threshold is related to what you assume about nucleation. Secondary nucleation is about breaking off fragments of ice off, so presumably the minimum size might relate to this process and might not necessarily be the same as the scale that might be expected from classical nucleation theory.
- L386-395: Of course, in geophysical contexts, this uncertainty is even worse.
- L415: $e$ and $r$ should be italicized.
- Around equation (25), perhaps link back to equations (3) and (4) somewhere. It's not immediately clear how great a range of uncertainty there is in Nu as Nu is a rather complex function of the parameters. Also the logarithm symbol should be in Roman font.
- L445: A more general issue is that the choices of parameters are not independent, but may trade off against each other. I think this could have been discussed more strongly at various points.
- L447 paragraph: the meaning of these symbols is defined quite a long way removed from this section, perhaps a bit more of a reminder of what they mean might help follow the paragraph.
- Figure 4: there seems to be a slope variation from w proportional to r to w proportional to $r^n$ where $n \approx 1/2$ which will reflect different dynamical regimes. The simplified approach (a_d) constant doesn't have this feature.
- Table 2: consider adding a column of parameter names/descriptors

- Table 3: Perhaps add the simplifications ($t\_s$,$a\_d=0$) in cases 1-2 more clearly (or in the caption).
- L517/L550: the sensitivity to initial conditions that you find is quite worrying for users of such models, as the initial conditions will be hard to know/control. I wondered if you had thought about what controls how long the ICs matter for?
- L572: the logic seems somewhat back-to-front here. It isn't the initial distribution that's key here, but rather what you assume about nucleation processes.
- L634: this might be true in the lab but not clear in the field
- L639: this is convincingly shown in this context (and is an important outcome), but might not be true in more complex situations where there is more complex evolution of the mean crystal size.
- L741: Rees Jones, D. W. (not Jones, D. W. R)

---

## Author Comment (AC1)

**Response to Mark Loewen :**

First of all, the authors would like to thank you for the constructive reviews, that helped us improve the quality of the manuscript. We have much appreciated your help in guaranteeing the accuracy and precision of assumptions and quantities mentioned in the text, as well as encouraging the clarity of the definitions and detailed explanations. We corrected the manuscript accordingly. Please find below our responses to your remarks.

Best regards,
Fabien Souillé, on behalf of the authors.

Figure: In all figures with multiple plots you should label the plots (a), (b) etc. to avoid confusion.
Absolutely. We added (a), (b) etc. the to plots.

74: Change "supposed to be" to "assumed to be".
As recommended, this was corrected.

191-193: Explain the reasoning for fixing the number of initial particles at zero for classes exceeding a radius threshold.
There has been various approaches in previous works concerning the choice of the initial distribution for particle sizes. Introducing a threshold r0 and considering it as uncertain allowed us to study the influence of this parameter since different authors used different r0 without really justifying why. To clarify this point, I added another reference in the text, and explained that our goal was to stay close to previous works and check if this parameter actually matters.

In practice we could have tried other methods such as log-normal initial distribution, but we lack data to support this modeling choice. This can for example be the object of future investigations to study the influence of the distribution shape itself on the uncertainty and sensitivity analysis.

262: Define UQ.
As recommended, we added the definition.

Sec. 3.3: I recommend this be moved to an appendix.
Our second referee, Dr David Rees Jones, also commented that this section should be summarized and moved to appendix. We accordingly moved most technical details of section 3.3 in appendix. It indeed simplifies the reading, thank you for this recommendation.

384: I agree the uncertainty in the initial volume fraction is very large.

500: In Fig. 5 time is plotted in seconds so referring to minutes in the text is inconsistent. Also please check the value of 0.1 C I think it is inaccurate.
Absolutely, thank you for this remark. We replaced minutes by seconds in the text, and 0.1 by the precise value of 0.097°C.

511: Are you referring to the median value of the time of maximum supercooling here?
Exactly. This was not clear in the text, I clarified it.

514: I think for experiments this argument is valid but I am not convinced this is true in rivers. There are very few reliable measurements of dissipation in rivers available and the uncertainty could easily be an order of magnitude or larger.

Thank you for raising this issue. We corrected this in the text by underlining that in the absence of data, we could still use approximations to estimate turbulent dissipation in rivers (such as equation (25), or more generally k-epsilon models in hydraulic codes). It might of course not be as reliable as measurements but still provide a rough idea of the order of magnitude for epsilon. In contrast, for initial concentration, we don't have any model to estimate it, which makes it more difficult to set in models.

518: Here you refer to "the initial distribution parameter" and it would be helpful to state also that these are C0 and r0 .

I added C0 and r0.

520: In Table 1 C0 and r0 are categorized as "Initial conditions" and here they are referred to as initial distribution. I find this confusing.

For the MSC model, the initial distribution is part of the initial condition of the system. One has to provide an initial volume fraction value for each class of radius and this is done via C0 and r0 in the presented methodology. But more generally, I wanted to stress that there is no particular reason to keep C0 and r0 and the presented methodology to set the initial distribution. We could use a log-normal distribution for example, which would be characterized by other parameters. I added precision on that.

521: You write "At the recovery" and later refer to a "recovery time" and these are both too vague. You define the "recovery phase" previously but this includes all time after steady state is reached. So clear terminology and clear definitions are required. In the same line you write "the parameters of secondary nucleation and flocculation processes". Please list all of these parameters here.

This was indeed not precise enough. I added a definition for the recovery time, and recovery phase in section 2.5. I also listed the parameters for secondary nucleation and flocculation.

523: How did you observe interactions between parameters?

In the diagrams of first order Sobol indices, the blank space separating the last First order Sobol index and 1 actually corresponds to the sum of all interactions (high order Sobol indices). So by seeing an increasing blank space in these figures, we can see that interactions play a important role. Another way to see the increase in interactions is to observe the evolution of Total Sobol indices in Appendix F, G, H and I. To clarify this in the manuscript, I added explanation and reference to Total Sobol figures. I also added more precision in Section 3 to explain what First order and total Sobol indices stand for.

Figure 8: Supercooling of -0.2 C is quite extreme so some comments on these values are required. In the right plot please also comment on the fact that for P95 the maximum supercooling is not reached even after 3000 s.

I added the following explanation in section 5.3 : this is due to the highest values of the buoyancy velocity combined with low thermal growth rate. High gravitational removal withdraw a large amount of the frazil volume fraction from the water and thus limits the amount of latent heat release to water that would increase its temperature. This combined action impacts both the time to maximum supercooling and the amount of supercooling.

546: List the parameters please.

Done.

552; Replace "coherent" with consistent?
I changed coherent to consistent.

554-555: Excellent point I agree.

559: Spelling changed to predicted.
Corrected.

560-562: This is a very limited discussion of Figure 10. Seems too brief - presumably there is more to discuss.
I added reference to similar observation made by Carstens (1966) and Ye et al. (2004) and discussed the figure more extensively.

571-573: Awkward wording, please rewrite this sentence.
I rephrased and hope this is clearer.

Figure 11; Add info to the caption to explain the symbols i.e., dots, error bars etc.
I added details in the caption.

582: I think you mean quantitative here not qualitative.
Exactly, corrected.

591-599: Excellent discussion here.
Thank you.

599: Define API.
I added the definition.

632: Replace relevant with suitable or promising?
Replaced by suitable.

640: I found it very interesting that you found that the turbulent dissipation rate plays a major role. Laboratory studies have found that the mean particle size varied with dissipation rate but the results are inconsistent. Can you use your model to examine this?
In the present work, we just looked at the total volume fraction, but the methodology could definitely be extended to the analysis of class volume fraction, the evolution of the output distribution or at least statistical moments of the output distribution such as the mean radius.
The MSC model could definitely be used to explore this by running simulations with different dissipation rates and analyse the evolution of the mean particle size.
And more generally, I think using Monte Carlo simulations with the MSC model could be used to help define new models for class interactions, like flocculation.

640-641: I do not agree that the dissipation rate is often appropriately quantified. Reliable measurements of dissipation rates in rivers are virtually non-existent and even in the lab it has not been accurately measured very often.
Thank you for raising this issue, also linked to your comment on line 514. I rephrased to reflect the idea that It could at least be modeled (even if it is not properly quantified).

645-646: You write "The long-term evolution of the system also showed increasing interactions between parameters, which can be explained by the balance in the physical

processes involved in class interactions". This was not clear to me since I do not think you explained this well in the paper.

Yes, this needed clarification.

From a computational standpoint, at steady state, the shape of the distribution (class volume fractions) is stable and the stability is due to the balance of thermal growth with secondary nucleation and flocculation. I think this equilibrium explained the observed interactions. I added an explanation in the text, as well as a reference to appendices for interactions estimation. Also, I rephrased « which can be » to « which could be » as it is more  a possible interpretation than a direct result. This could be more precisely quantified by comparison to simulations where one of the terms (either flocculation or secondary nucleation) is set to 0. Using class volume fraction as model outputs could also help to better understand this.

652-653: Your conclusion regarding the rise velocity is a very significant result – well done!

Thank you. I was quite surprised by this result. I was expecting an influence, but not that important compared to other parameters. Now my dream would be to have unlimited data with clear vertical distribution of frazil so we can validate non-well mixed numerical models including buoyancy velocity.

---

## Author Comment (AC2)

**Response to David Rees Jones :**

First of all, the authors would like to thank you for the constructive reviews, that helped us improve the quality of the manuscript,. In particular, we appreciated your remarks about the clarity of notations, the justification of numerical choices, and the clarity of modelling assumptions, and have made changes to the text accordingly. Please find below our responses to your remarks.

Best regards,
Fabien Souillé, on behalf of the authors.

- Code/data: I was not sure whether the code developed/used in this study was already available and if so where? The submission guidelines suggest: Authors are encouraged to deposit software, algorithms, and model code in FAIR-aligned repositories/archives whenever possible. These research outputs are then cited in the manuscript using the received DOI and included in the reference list. The manuscript must then include a section entitled "Code availability" or, in the case of data and code, "Code and data availability".
  The code developed for this study is not directly available. However, most of it is implemented and available in the open-source TELEMAC-MASCARET software within the module KHIONE dedicated to ice modeling. This includes both single- and multiple-class models. The only difference is that the semi-implicit theta scheme with the linear system presented is this publication is not incorporated in KHIONE yet.

- L18: Could mention geophysical contexts like plumes of Ice Shelf Water under floating ice sheets.
  Thank you for this suggestions. I have added a sentence and references on the topic of ice shelf water plumes.

- L48: Sentence probably needs splitting.
  As recommended, I split the sentence.

- L50: Perhaps add or substitute a geophysical example (e.g. from seismology)
  Thank you for pointing out seismology, because there is some really insightful literature in this field especially on how to deal with models uncertainties. I have changed the sentence to include references to both geophysical and environmental modeling.

- L56: A different kind of thing, but probabilistic methods are sometimes used in processing observational data (Frazer, E. K., Langhorne, P. J., Leonard, G. H., Robinson, N. J., & Schumayer, D. (2020). Observations of the size distribution of frazil ice in an Ice Shelf Water plume. Geophysical Research Letters, 47, https://doi.org/10.1029/2020GL090498)
  Thank you for this very interesting reference, which I have added to the introduction.

- L74: The choice of e to denote thickness was a bit confusing as the letter is usually reserved for Euler's number. I'm not sure if there is a particular precedent or motivation. Perhaps e related to edge length? Similarly, the definition of a above equation (2) might be somewhat confusing (as it denotes only part of the surface

area, not the whole). If there is a source of both of these choices come from, it would be good to mention it.

The choice of notation for the thickness was difficult to make since « t » and « h » were already used. We chose e because it stands for « épaisseur » in French (literal translation of thickness), but this was probably not the best choice as you pointed out. So I've changed the notation to: lambda.

For the definition of «a»: in previous works, their is a common assumption that frazil crystals grow from their peripheral area (which we denoted a), but Holland and Feltham (2005) proposed in their model that the crystals melt from their whole surface (denoted s). In the present study, we kept similar assumptions to be consistent with previous modeling works. L55 we expose this hypothesis : «We suppose that frazil crystals grow from their peripheral area $a_i$ but melt from their surface $s_i$ (Holland and Feltham, 2005)».

- L92: not very clear what delta_T refers to at this stage (I see you come back and discuss it later, so it would be sensible to add some cross references and/or consider consolidating the discussion)
  DeltaT was here used as a generic notation to refer to either the radius or the thickness, depending on the choice that is made for the scaling. It is actually similar to f/H in your paper (D. W. Rees Jones and A. J. Wells: Frazil-ice growth rate and dynamics). f = f2 = 1 <=> DeltaT = thickness and f = f3 = H/R <=> DeltaT = radius. I have added a sentence to explain that more clearly.

- L108-109: It seems to be assumed that seeding and secondary nucleation produce crystals of the same size. This might be as good as any other assumption but could be spelled out more clearly
  Absolutely, this was not clearly mentioned. I added a sentence to state this assumption.

- L118: while relying on this precedent is fine, I think the main issue is that the flocculation rate in this form is independent of the number of the number of crystals. The nucleation term written (2) in equation (1) will be proportional to the square of the number of crystals while the flocculation term written (3) will be linear in the number of crystals. [I would think that both these terms arise from the same type of processes and should both be quadratic.]
  Yes I agree with your reasoning on the Flocculation term. I've added more details on this in the text. I think that there is really a gap in terms of modeling on flocculation, If we compare for example with what is done in sedimentology. As a perspective for this work on multiple-size-class models, It would be interesting to study the sensitivity to different formulations for both nucleation and flocculation, and see what formulations fit better to the observed evolution of distributions.
  But I think the difficulty here would be that distributions only gives us an idea of the global balance between all processes combined. Validating a single process independently from the others would be a challenge.

- L180-188: the reference to Appendix A could have been put nearer the start of the paragraph? The whole of this paragraph was relatively technical and could have been move the appendix, perhaps. It would have been good to have a brief explanation of how this scheme was chosen (especially because other studies didn't necessarily use the same method).
  As suggested, I moved most of the details on the numerical scheme to appendix A.

Concerning the scheme choice, the stability condition imposes to set time steps that are relatively small. This condition is less restrictive with a semi-implicit method which allows larger time steps. This was the reason why we chose the semi-implicit method. Indeed, the used Monte Carlo sampling for the uncertainty investigation implies thousands of simulations to run, and the scheme choice was in this context crucial to lower the cost of the study. The semi-implicit approach was also used by Wang and Doering (2005). For other studies, the time scheme is not always described, so my guess is that Euler forward is more often used since it is easier to implement, despite the fact that the time step constraint can be very limiting.

- Figure 1: I found the graphs quite cluttered and hard to read. Could experiment with different colour schemes, larger figure panels (there was quite a bit of white space), perhaps one/two less data series. Perhaps the left panel would have been more useful plotted at some later time instead (e..g t=300s).
  To improve the visual clarity of this Figure, I removed m=500 and m=1000. I also added the number of crystals per class at t=300s on left plot.

- Table 1: I spotted n_max here but missed where it was discussed in the main text (I saw it in L160 but not discussed in paragraph starting L105). It is quite a significant fudge factor so needs discussing somewhere in words when the notation is first introduced.
  The parameter is introduced in section 2.1 when secondary nucleation is presented. «ñ = max(N, n max ) is the average number of particles per unit volume that take part in the collisions, and n_max is a fitting parameter controlling the efficiency of the collision breeding». Perhaps this needed a reminder in L160, that I added.

- Section 3: I felt this could be moved to an appendix/supplement. It wasn't clear that anything was particular to this manuscript. Perhaps a one paragraph summary could go at the start of what is currently section 4.
  The other referre, Dr Mark Loewen, had the same comment. I moved most technical details of section 3.3 in an appendix. However I do prefer to keep a section 3 to explain the methodology that was followed, I think it's important to understand part 4 and 5, especially for readers that are not familiar with statistical methods.

- L360: Presumably the minimum threshold is related to what you assume about nucleation. Secondary nucleation is about breaking off fragments of ice off, so presumably the minimum size might relate to this process and might not necessarily be the same as the scale that might be expected from classical nucleation theory.
  Yes, that is true, thank you for pointing it out. Secondary nucleation may feed a specific radius, or a selected range of radii, which are not necessarily the smallest radius class. That way, secondary nucleation would be independent from the discretization of the radius space but at the cost of new parameters. In the current study, we  assumed that secondary nucleation only feeds the smallest radius class in the model, which makes the minimum radius connected to secondary nucleation. This assumption needed to be stated out more clearly in the text, and we adapted the manuscript accordingly.

- L386-395: Of course, in geophysical contexts, this uncertainty is even worse.

- L415: e and r should be italicized.
  This was corrected.

- Around equation (25), perhaps link back to equations (3) and (4) somewhere. It's not immediately clear how great a range of uncertainty there is in Nu as Nu is a rather complex function of the parameters. Also the logarithm symbol should be in Roman font.
  As recommended, I added reference to equations (3) and (4) and used Roman font for the logarithm symbol.
  During our investigations, we also tested Nu as an uncertain parameter initially. To estimate the uncertainty bounds and PDF of Nu we propagated the PDF of turbulent parameters and radius through equations (3) and (4) and obtained interesting results for the PDF of Nu: it was multi-modal, and the values ranged from approximately 1 to 25. However, we decided not to use this approach since Nu can't be considered independent from the radius. Considering it as uncertain may need more sophisticated modeling of probabilistic dependency and can be a good perspective (this is actually linked to your following remark). Another challenge may concern the discontinuities in functions (3) and (4).

- L445: A more general issue is that the choices of parameters are not independent, but may trade off against each other. I think this could have been discussed more strongly at various points.
  Yes, we tried to make choices of parameters to avoid modeling the probabilistic dependency for this first investigation. For example, we decided not to take Nu as a parameter, but selected turbulent parameters instead. But I agree that underlying dependencies remain. To clarify this, I have added more elements on dependency in section 4 and put more emphasis on that in the «limits and perspectives» section.

- L447 paragraph: the meaning of these symbols is defined quite a long way removed from this section, perhaps a bit more of a reminder of what they mean might help follow the paragraph.
  I have added a reminder for the description of these parameters.

- Figure 4: there seems to be a slope variation from w proportional to r to w proportional to r^n where n \approx 1/2 which will reflect different dynamical regimes. The simplified approach (a_d) constant doesn't have this feature.
  Yes, that's true. If we look at the data on which most of the laws were fitted, there is a significant scattering for small radius. The slope can therefore vary a lot for small radius. We kept the law of w that represented the median behavior, so that adding a constant envelope actually contains the majority of the scattered data.
  To take the change of slope into account, an option would be to model the dependency of w to r. Another idea would be to take a simple but generic enough law, like w=a*r^n and model the uncertainty of both parameters a and n. But defining proper distribution for a and n is not straightforward.

- Table 2: consider adding a column of parameter names/descriptors
  I added a column of description like in Table 1.

- Table 3: Perhaps add the simplifications (t_s, a_d=0) in cases 1-2 more clearly (or in the caption).
  I added t_s=0, a_d=0 in the caption.

- L517/L550: the sensitivity to initial conditions that you find is quite worrying for users of such models, as the initial conditions will be hard to know/control. I wondered if you had thought about what controls how long the ICs matter for?
  It seems that the IC is very influential prior to maximum supercooling. So I don't think the model is actually good at predicting the maximum supercooling point, unless it has been calibrated on very similar setups to the one modeled. With that said, what surprised me the most when I started working on these models is the steady state that does not depend on thermal growth parameters but only on the heat sink rate (in the absence of buoyancy removal). So IC doesn't really matter to estimate the amount of frazil past maximum supercooling. Of course when adding buoyancy, the steady state depends on the equilibrium between the heat sink rate and buoyancy removal, in which case I don't think the well mixed models are precise enough to be predictive. However, this limitation doesn't come from IC but rather from buoyancy velocity.

- L572: the logic seems somewhat back-to-front here. It isn't the initial distribution that's key here, but rather what you assume about nucleation processes.
  I agree  on the fact that the assumption of nucleation feeding only  the first class has an impact on the results. However, I think that there may be other reasons why we observe different behaviors by changing minimum radius (2b) and initial distribution (2c). In fact the volume growth rate is bigger for small classes so, the more initial concentration is attributed to small classes, the faster the maximum supercooling is reached. This is what we observe in case (2c). In this section I wanted to stress out that it is a combination of both nucleation and thermal growth processes that makes the choice of r_min and initial distribution important.
  I rephrased part of this section to reflect that idea.

- L634: this might be true in the lab but not clear in the field
  I modified the text to precise that.

- L639: this is convincingly shown in this context (and is an important outcome), but might not be true in more complex situations where there is more complex evolution of the mean crystal size.
  I agree that the SSC model might not be representative of the complex situations observed in reality.  The fact that the mean radius is the most influential here is only a property of the SSC model, since using it requires choosing a mean radius, and does not necessarily reflect reality.

- L741: Rees Jones, D. W. (not Jones, D. W. R)
  It was corrected, I apologize for the mistake.